# SAMPLE EFFICIENT POLICY GRADIENT METHODS WITH RECURSIVE VARIANCE REDUCTION

**Pan Xu, Felicia Gao, Quanquan Gu**
Department of Computer Science
University of California, Los Angeles
Los Angeles, CA 90094, USA
panxu@cs.ucla.edu, fxgao1160@engineering.ucla.edu, qgu@cs.ucla.edu

## ABSTRACT

Improving the sample efficiency in reinforcement learning has been a long-standing research problem. In this work, we aim to reduce the sample complexity of existing policy gradient methods. We propose a novel policy gradient algorithm called SRVR-PG, which only requires $O(1/\epsilon^{3/2})$[1] episodes to find an $\epsilon$-approximate stationary point of the nonconcave performance function $J(\boldsymbol{\theta})$ (i.e., $\boldsymbol{\theta}$ such that $\|\nabla J(\boldsymbol{\theta})\|_2^2 \le \epsilon$). This sample complexity improves the existing result $O(1/\epsilon^{5/3})$ for stochastic variance reduced policy gradient algorithms by a factor of $O(1/\epsilon^{1/6})$. In addition, we also propose a variant of SRVR-PG with parameter exploration, which explores the initial policy parameter from a prior probability distribution. We conduct numerical experiments on classic control problems in reinforcement learning to validate the performance of our proposed algorithms.

## 1 INTRODUCTION

Reinforcement learning (RL) (Sutton & Barto, 2018) has received significant success in solving various complex problems such as learning robotic motion skills (Levine et al., 2015), autonomous driving (Shalev-Shwartz et al., 2016) and Go game (Silver et al., 2017), where the agent progressively interacts with the environment in order to learn a good policy to solve the task. In RL, the agent makes its decision by choosing the action based on the current state and the historical rewards it has received so far. After performing the chosen action, the agent's state will change according to some transition probability model and a new reward would be revealed to the agent by the environment based on the action and new state. Then the agent continues to choose the next action until it reaches a terminal state. The aim of the agent is to maximize its expected cumulative rewards. Therefore, the pivotal problem in RL is to find a good policy which is a function that maps the state space to the action space and thus informs the agent which action to take at each state. To optimize the agent's policy in the high dimensional continuous action space, the most popular approach is the policy gradient method (Sutton et al., 2000) that parameterizes the policy by an unknown parameter $\boldsymbol{\theta} \in \mathbb{R}^d$ and directly optimizes the policy by finding the optimal $\boldsymbol{\theta}$. The objective function $J(\boldsymbol{\theta})$ is chosen to be the performance function, which is the expected return under a specific policy and is usually non-concave. Our goal is to maximize the value of $J(\boldsymbol{\theta})$ by finding a stationary point $\boldsymbol{\theta}^*$ such that $\|\nabla J(\boldsymbol{\theta}^*)\|_2 = 0$ using gradient based algorithms.

Due to the expectation in the definition of $J(\boldsymbol{\theta})$, it is usually infeasible to compute the gradient exactly. In practice, one often uses stochastic gradient estimators such as REINFORCE (Williams, 1992), PGT (Sutton et al., 2000) and GPOMDP (Baxter & Bartlett, 2001) to approximate the gradient of the expected return based on a batch of sampled trajectories. However, this approximation will introduce additional variance and slow down the convergence of policy gradient, which thus requires a huge amount of trajectories to find a good policy. Theoretically, these stochastic gradient (SG) based algorithms require $O(1/\epsilon^2)$ trajectories (Robbins & Monro, 1951) to find an $\epsilon$-approximate stationary point such that $\mathbb{E}[\|\nabla J(\boldsymbol{\theta})\|_2^2] \le \epsilon$. In order to reduce the variance of policy gradient algorithms, Papini et al. (2018) proposed a stochastic variance-reduced policy gradient (SVRPG)

---

[1]$O(\cdot)$ notation hides constant factors.

Table 1: Comparison on sample complexities of different algorithms to achieve $\|\nabla J(\boldsymbol{\theta})\|_2^2 \leq \epsilon$.

| Algorithms | Complexity |
|---|---|
| REINFORCE (Williams, 1992) | $O(1/\epsilon^2)$ |
| PGT (Sutton et al., 2000) | $O(1/\epsilon^2)$ |
| GPOMDP (Baxter & Bartlett, 2001) | $O(1/\epsilon^2)$ |
| SVRPG (Papini et al., 2018) | $O(1/\epsilon^2)$ |
| SVRPG (Xu et al., 2019) | $O(1/\epsilon^{5/3})$ |
| SRVR-PG (This paper) | $O(1/\epsilon^{3/2})$ |

algorithm by borrowing the idea from the stochastic variance reduced gradient (SVRG) (Johnson & Zhang, 2013; Allen-Zhu & Hazan, 2016; Reddi et al., 2016a) in stochastic optimization. The key idea is to use a so-called semi-stochastic gradient to replace the stochastic gradient used in SG methods. The semi-stochastic gradient combines the stochastic gradient in the current iterate with a snapshot of stochastic gradient stored in an early iterate which is called a reference iterate. In practice, SVRPG saves computation on trajectories and improves the performance of SG based policy gradient methods. Papini et al. (2018) also proved that SVRPG converges to an $\epsilon$-approximate stationary point $\boldsymbol{\theta}$ of the nonconcave performance function $J(\boldsymbol{\theta})$ with $\mathbb{E}[\|\nabla J(\boldsymbol{\theta})\|_2^2] \leq \epsilon$ after $O(1/\epsilon^2)$ trajectories, which seems to have the same sample complexity as SG based methods. Recently, the sample complexity of SVRPG has been improved to $O(1/\epsilon^{5/3})$ by a refined analysis (Xu et al., 2019), which theoretically justifies the advantage of SVRPG over SG based methods.

This paper continues on this line of research. We propose a Stochastic Recursive Variance Reduced Policy Gradient algorithm (SRVR-PG), which provably improves the sample complexity of SVRPG. At the core of our proposed algorithm is a recursive semi-stochastic policy gradient inspired from the stochastic path-integrated differential estimator (Fang et al., 2018), which accumulates all the stochastic gradients from different iterates to reduce the variance. We prove that SRVR-PG only takes $O(1/\epsilon^{3/2})$ trajectories to converge to an $\epsilon$-approximate stationary point $\boldsymbol{\theta}$ of the performance function, i.e., $\mathbb{E}[\|\nabla J(\boldsymbol{\theta})\|_2^2] \leq \epsilon$. We summarize the comparison of SRVR-PG with existing policy gradient methods in terms of sample complexity in Table 1. Evidently, the sample complexity of SRVR-PG is lower than that of REINFORCE, PGT and GPOMDP by a factor of $O(1/\epsilon^{1/2})$, and is lower than that of SVRPG (Xu et al., 2019) by a factor of $O(1/\epsilon^{1/6})$.

In addition, we integrate our algorithm with parameter-based exploration (PGPE) method (Sehnke et al., 2008; 2010), and propose a SRVR-PG-PE algorithm which directly optimizes the prior probability distribution of the policy parameter $\boldsymbol{\theta}$ instead of finding the best value. The proposed SRVR-PG-PE enjoys the same trajectory complexity as SRVR-PG and performs even better in some applications due to its additional exploration over the parameter space. Our experimental results on classical control tasks in reinforcement learning demonstrate the superior performance of the proposed SRVR-PG and SRVR-PG-PE algorithms and verify our theoretical analysis.

## 1.1 ADDITIONAL RELATED WORK

We briefly review additional relevant work to ours with a focus on policy gradient based methods. For other RL methods such as value based (Watkins & Dayan, 1992; Mnih et al., 2015) and actor-critic (Konda & Tsitsiklis, 2000; Peters & Schaal, 2008a; Silver et al., 2014) methods, we refer the reader to Peters & Schaal (2008b); Kober et al. (2013); Sutton & Barto (2018) for a complete review.

To reduce the variance of policy gradient methods, early works have introduced unbiased baseline functions (Baxter & Bartlett, 2001; Greensmith et al., 2004; Peters & Schaal, 2008b) to reduce the variance, which can be constant, time-dependent or state-dependent. Schulman et al. (2015b) proposed the generalized advantage estimation (GAE) to explore the trade-off between bias and variance of policy gradient. Recently, action-dependent baselines are also used in Tucker et al. (2018); Wu et al. (2018) which introduces bias but reduces variance at the same time. Sehnke et al. (2008; 2010) proposed policy gradient with parameter-based exploration (PGPE) that explores in the parameter space. It has been shown that PGPE enjoys a much smaller variance (Zhao et al.,

2011). The Stein variational policy gradient method is proposed in Liu et al. (2017). See Peters & Schaal (2008b); Deisenroth et al. (2013); Li (2017) for a more detailed survey on policy gradient.

Stochastic variance reduced gradient techniques such as SVRG (Johnson & Zhang, 2013; Xiao & Zhang, 2014), batching SVRG (Harikandeh et al., 2015), SAGA (Defazio et al., 2014) and SARAH (Nguyen et al., 2017) were first developed in stochastic convex optimization. When the objective function is nonconvex (or nonconcave for maximization problems), nonconvex SVRG (Allen-Zhu & Hazan, 2016; Reddi et al., 2016a) and SCSG (Lei et al., 2017; Li & Li, 2018) were proposed and proved to converge to a first-order stationary point faster than vanilla SGD (Robbins & Monro, 1951) with no variance reduction. The state-of-the-art stochastic variance reduced gradient methods for nonconvex functions are the SNVRG (Zhou et al., 2018) and SPIDER (Fang et al., 2018) algorithms, which have been proved to achieve near optimal convergence rate for smooth functions.

There are yet not many papers studying variance reduced gradient techniques in RL. Du et al. (2017) first applied SVRG in policy evaluation for a fixed policy. Xu et al. (2017) introduced SVRG into trust region policy optimization for model-free policy gradient and showed that the resulting algorithm SVRPO is more sample efficient than TRPO. Yuan et al. (2019) further applied the techniques in SARAH (Nguyen et al., 2017) and SPIDER (Fang et al., 2018) to TRPO (Schulman et al., 2015a). However, no analysis on sample complexity (i.e., number of trajectories required) was provided in the aforementioned papers (Xu et al., 2017; Yuan et al., 2019). We note that a recent work by Shen et al. (2019) proposed a Hessian aided policy gradient (HAPG) algorithm that converges to the stationary point of the performance function within $O(H^2/\epsilon^{3/2})$ trajectories, which is worse than our result by a factor of $O(H^2)$ where $H$ is the horizon length of the environment. Moreover, they need additional samples to approximate the Hessian vector product, and cannot handle the policy in a constrained parameter space. Another related work pointed out by the anonymous reviewer is Yang & Zhang (2019), which extended the stochastic mirror descent algorithm (Ghadimi et al., 2016) in the optimization field to policy gradient methods and achieved $O(H^2/\epsilon^2)$ sample complexity. After the ICLR conference submission deadline, Yang & Zhang (2019) revised their paper by adding a new variance reduction algorithm that achieves $O(H^2/\epsilon^{3/2})$ sample complexity, which is also worse than our result by a factor of $O(H^2)$.

Apart from the convergence analysis of the general nonconcave performance functions, there has emerged a line of work (Cai et al., 2019; Liu et al., 2019; Yang et al., 2019; Wang et al., 2019) that studies the global convergence of (proximal/trust-region) policy optimization with neural network function approximation, which applies the theory of overparameterized neural networks (Du et al., 2019b;a; Allen-Zhu et al., 2019; Zou et al., 2019; Cao & Gu, 2019) to reinforcement learning.

**Notation** $\|\mathbf{v}\|_2$ denotes the Euclidean norm of a vector $\mathbf{v} \in \mathbb{R}^d$ and $\|\mathbf{A}\|_2$ denotes the spectral norm of a matrix $\mathbf{A} \in \mathbb{R}^{d \times d}$. We write $a_n = O(b_n)$ if $a_n \leq Cb_n$ for some constant $C > 0$. The Dirac delta function $\delta(x)$ satisfies $\delta(0) = +\infty$ and $\delta(x) = 0$ if $x \neq 0$. Note that $\delta(x)$ satisfies $\int_{-\infty}^{+\infty} \delta(x)\mathrm{d}x = 1$. For any $\alpha > 0$, we define the Rényi divergence (Rényi et al., 1961) between distributions $P$ and $Q$ as

$$D_\alpha(P||Q) = \frac{1}{\alpha - 1}\log_2 \int_x P(x)\left(\frac{P(x)}{Q(x)}\right)^{\alpha - 1}\mathrm{d}x,$$

which is non-negative for all $\alpha > 0$. The exponentiated Rényi divergence is $d_\alpha(P||Q) = 2^{D_\alpha(P||Q)}$.

## 2    BACKGROUNDS ON POLICY GRADIENT

**Markov Decision Process:** A discrete-time Markov Decision Process (MDP) is a tuple $\mathcal{M} = \{\mathcal{S}, \mathcal{A}, \mathcal{P}, r, \gamma, \rho\}$. $\mathcal{S}$ and $\mathcal{A}$ are the state and action spaces respectively. $\mathcal{P}(s'|s, a)$ is the transition probability of transiting to state $s'$ after taking action $a$ at state $s$. Function $r(s, a) : \mathcal{S} \times \mathcal{A} \to [-R, R]$ emits a bounded reward after the agent takes action $a$ at state $s$, where $R > 0$ is a constant. $\gamma \in (0, 1)$ is the discount factor. $\rho$ is the distribution of the starting state. A policy at state $s$ is a probability function $\pi(a|s)$ over action space $\mathcal{A}$. In episodic tasks, following any stationary policy, the agent can observe and collect a sequence of state-action pairs $\tau = \{s_0, a_0, s_1, a_1, \ldots, s_{H-1}, a_{H-1}, s_H\}$, which is called a trajectory or episode. $H$ is called the trajectory horizon or episode length. In practice, we can set $H$ to be the maximum value among all

the actual trajectory horizons we have collected. The sample return over one trajectory $\tau$ is defined as the discounted cumulative reward $\mathcal{R}(\tau) = \sum_{h=0}^{H-1} \gamma^h r(s_h, a_h)$.

**Policy Gradient:** Suppose the policy, denoted by $\pi_{\boldsymbol{\theta}}$, is parameterized by an unknown parameter $\boldsymbol{\theta} \in \mathbb{R}^d$. We denote the trajectory distribution induced by policy $\pi_{\boldsymbol{\theta}}$ as $p(\tau|\boldsymbol{\theta})$. Then

$$p(\tau|\boldsymbol{\theta}) = \rho(s_0) \prod_{h=0}^{H-1} \pi_{\boldsymbol{\theta}}(a_h|s_h) P(s_{h+1}|s_h, a_h). \tag{2.1}$$

We define the expected return under policy $\pi_{\boldsymbol{\theta}}$ as $J(\boldsymbol{\theta}) = \mathbb{E}_{\tau \sim p(\cdot|\boldsymbol{\theta})}[\mathcal{R}(\tau)|\mathcal{M}]$, which is also called the performance function. To maximize the performance function, we can update the policy parameter $\boldsymbol{\theta}$ by iteratively running gradient ascent based algorithms, i.e., $\boldsymbol{\theta}_{k+1} = \boldsymbol{\theta}_k + \eta \nabla_{\boldsymbol{\theta}} J(\boldsymbol{\theta}_k)$, where $\eta > 0$ is the step size and the gradient $\nabla_{\boldsymbol{\theta}} J(\boldsymbol{\theta})$ is derived as follows:

$$\nabla_{\boldsymbol{\theta}} J(\boldsymbol{\theta}) = \int_{\tau} \mathcal{R}(\tau) \nabla_{\boldsymbol{\theta}} p(\tau|\boldsymbol{\theta}) \mathrm{d}\tau = \int_{\tau} \mathcal{R}(\tau)(\nabla_{\boldsymbol{\theta}} p(\tau|\boldsymbol{\theta})/p(\tau|\boldsymbol{\theta})) p(\tau|\boldsymbol{\theta}) \mathrm{d}\tau$$
$$= \mathbb{E}_{\tau \sim p(\cdot|\boldsymbol{\theta})}[\nabla_{\boldsymbol{\theta}} \log p(\tau|\boldsymbol{\theta}) \mathcal{R}(\tau)|\mathcal{M}]. \tag{2.2}$$

However, it is intractable to calculate the exact gradient in (2.2) since the trajectory distribution $p(\tau|\boldsymbol{\theta})$ is unknown. In practice, policy gradient algorithm samples a batch of trajectories $\{\tau_i\}_{i=1}^N$ to approximate the exact gradient based on the sample average over all sampled trajectories:

$$\widehat{\nabla}_{\boldsymbol{\theta}} J(\boldsymbol{\theta}) = \frac{1}{N} \sum_{i=1}^N \nabla_{\boldsymbol{\theta}} \log p(\tau_i|\boldsymbol{\theta}) \mathcal{R}(\tau_i). \tag{2.3}$$

At the $k$-th iteration, the policy is then updated by $\boldsymbol{\theta}_{k+1} = \boldsymbol{\theta}_k + \eta \widehat{\nabla}_{\boldsymbol{\theta}} J(\boldsymbol{\theta}_k)$. According to (2.1), we know that $\nabla_{\boldsymbol{\theta}} \log p(\tau_i|\boldsymbol{\theta})$ is independent of the transition probability matrix $P$. Recall the definition of $\mathcal{R}(\tau)$, we can rewrite the approximate gradient as follows

$$\widehat{\nabla}_{\boldsymbol{\theta}} J(\boldsymbol{\theta}) = \frac{1}{N} \sum_{i=1}^N \left( \sum_{h=0}^{H-1} \nabla_{\boldsymbol{\theta}} \log \pi_{\boldsymbol{\theta}}(a_h^i|s_h^i) \right) \left( \sum_{h=0}^{H-1} \gamma^h r(s_h^i, a_h^i) \right)$$
$$\stackrel{\text{def}}{=} \frac{1}{N} \sum_{i=1}^N g(\tau_i|\boldsymbol{\theta}), \tag{2.4}$$

where $\tau_i = \{s_0^i, a_0^i, s_1^i, a_1^i, \ldots, s_{H-1}^i, a_{H-1}^i, s_H^i\}$ for all $i = 1, \ldots, N$ and $g(\tau_i|\boldsymbol{\theta})$ is an unbiased gradient estimator computed based on the $i$-th trajectory $\tau_i$. The gradient estimator in (2.4) is based on the likelihood ratio methods and is often referred to as the REINFORCE gradient estimator (Williams, 1992). Since $\mathbb{E}[\nabla_{\boldsymbol{\theta}} \log \pi_{\boldsymbol{\theta}}(a|s)] = 0$, we can add any constant baseline $b_t$ to the reward that is independent of the current action and the gradient estimator still remains unbiased. With the observation that future actions do not depend on past rewards, another famous policy gradient theorem (PGT) estimator (Sutton et al., 2000) removes the rewards from previous states:

$$g(\tau_i|\boldsymbol{\theta}) = \sum_{h=0}^{H-1} \nabla_{\boldsymbol{\theta}} \log \pi_{\boldsymbol{\theta}}(a_h^i|s_h^i) \left( \sum_{t=h}^{H-1} \gamma^t r(s_t^i, a_t^i) - b_t \right), \tag{2.5}$$

where $b_t$ is a constant baseline. It has been shown (Peters & Schaal, 2008b) that the PGT estimator is equivalent to the commonly used GPOMDP estimator (Baxter & Bartlett, 2001) defined as follows:

$$g(\tau_i|\boldsymbol{\theta}) = \sum_{h=0}^{H-1} \left( \sum_{t=0}^h \nabla_{\boldsymbol{\theta}} \log \pi_{\boldsymbol{\theta}}(a_t^i|s_t^i) \right) \left( \gamma^h r(s_h^i, a_h^i) - b_h \right). \tag{2.6}$$

All the three gradient estimators mentioned above are unbiased (Peters & Schaal, 2008b). It has been proved that the variance of the PGT/GPOMDP estimator is independent of horizon $H$ while the variance of REINFORCE depends on $H$ polynomially (Zhao et al., 2011; Pirotta et al., 2013). Therefore, we will focus on the PGT/GPOMDP estimator in this paper and refer to them interchangeably due to their equivalence.

## 3 THE PROPOSED ALGORITHM

The approximation in (2.3) using a batch of trajectories often causes a high variance in practice. In this section, we propose a novel variance reduced policy gradient algorithm called stochastic recursive variance reduced policy gradient (SRVR-PG), which is displayed in Algorithm 1. Our SRVR-PG algorithm consists of $S$ epochs. In the initialization, we set the parameter of a reference policy to be $\widetilde{\boldsymbol{\theta}}^0 = \boldsymbol{\theta}_0$. At the beginning of the $s$-th epoch, where $s = 0, \ldots, S - 1$, we set the initial policy parameter $\boldsymbol{\theta}_0^{s+1}$ to be the same as that of the reference policy $\widetilde{\boldsymbol{\theta}}^s$. The algorithm then samples $N$ episodes $\{\tau_i\}_{i=1}^N$ from the reference policy $\pi_{\widetilde{\boldsymbol{\theta}}^s}$ to compute a gradient estimator $\mathbf{v}_0^s = 1/N \sum_{i=1}^N g(\tau_i|\widetilde{\boldsymbol{\theta}}^s)$, where $g(\tau_i|\widetilde{\boldsymbol{\theta}}^s)$ is the PGT/GPOMDP estimator. Then the policy is immediately update as in Line 6 of Algorithm 1.

Within the epoch, at the $t$-th iteration, SRVR-PG samples $B$ episodes $\{\tau_j\}_{j=1}^B$ based on the current policy $\pi_{\boldsymbol{\theta}_t^{s+1}}$. We define the following recursive semi-stochastic gradient estimator:

$$\mathbf{v}_t^{s+1} = \frac{1}{B} \sum_{j=1}^B g(\tau_j|\boldsymbol{\theta}_t^{s+1}) - \frac{1}{B} \sum_{j=1}^B g_\omega(\tau_j|\boldsymbol{\theta}_{t-1}^{s+1}) + \mathbf{v}_{t-1}^{s+1}, \tag{3.1}$$

where the first term is a stochastic gradient based on $B$ episodes sampled from the current policy, and the second term is a stochastic gradient defined based on the *step-wise important weight* between the current policy $\pi_{\boldsymbol{\theta}_t^{s+1}}$ and the reference policy $\pi_{\widetilde{\boldsymbol{\theta}}^s}$. Take the GPOMDP estimator for example, for a behavior policy $\pi_{\boldsymbol{\theta}_1}$ and a target policy $\pi_{\boldsymbol{\theta}_2}$, the step-wise importance weighted estimator is defined as follows

$$g_\omega(\tau_j|\boldsymbol{\theta}_1) = \sum_{h=0}^{H-1} \omega_{0:h}(\tau|\boldsymbol{\theta}_2, \boldsymbol{\theta}_1) \left( \sum_{t=0}^h \nabla_{\boldsymbol{\theta}_2} \log \pi_{\boldsymbol{\theta}_2}(a_t^j|s_t^j) \right) \gamma^h r(s_h^j, a_h^j), \tag{3.2}$$

where $\omega_{0:h}(\tau|\boldsymbol{\theta}_2, \boldsymbol{\theta}_1) = \prod_{h'=0}^h \pi_{\boldsymbol{\theta}_2}(a_h|s_h)/\pi_{\boldsymbol{\theta}_1}(a_h|s_h)$ is the importance weight from $p(\tau_h|\boldsymbol{\theta}_t^{s+1})$ to $p(\tau_h|\boldsymbol{\theta}_{t-1}^{s+1})$ and $\tau_h$ is a truncated trajectory $\{(a_t, s_t)\}_{t=0}^h$ from the full trajectory $\tau$. It is easy to verify that $\mathbb{E}_{\tau \sim p(\tau|\boldsymbol{\theta}_1)}[g_\omega(\tau_j|\boldsymbol{\theta}_1)] = \mathbb{E}_{\tau \sim p(\tau|\boldsymbol{\theta}_2)}[g(\tau|\boldsymbol{\theta}_2)]$.

The difference between the last two terms in (3.1) can be viewed as a control variate to reduce the variance of the stochastic gradient. In many practical applications, the policy parameter space is a subset of $\mathbb{R}^d$, i.e., $\boldsymbol{\theta} \in \boldsymbol{\Theta}$ with $\boldsymbol{\Theta} \subseteq \mathbb{R}^d$ being a convex set. In this case, we need to project the updated policy parameter onto the constraint set. Base on the semi-stochastic gradient (3.1), we can update the policy parameter using projected gradient ascent along the direction of $\mathbf{v}_t^{s+1}$: $\boldsymbol{\theta}_{t+1}^{s+1} = \mathcal{P}_{\boldsymbol{\Theta}}(\boldsymbol{\theta}_t^{s+1} + \eta \mathbf{v}_t^{s+1})$, where $\eta > 0$ is the step size and the projection operator associated with $\boldsymbol{\Theta}$ is defined as

$$\mathcal{P}_{\boldsymbol{\Theta}}(\boldsymbol{\theta}) = \operatorname*{argmin}_{\mathbf{u} \in \boldsymbol{\Theta}} \|\boldsymbol{\theta} - \mathbf{u}\|_2^2 = \operatorname*{argmin}_{\mathbf{u} \in \mathbb{R}^d} \left\{ \mathbb{1}_{\boldsymbol{\Theta}}(\mathbf{u}) + \frac{1}{2\eta} \|\boldsymbol{\theta} - \mathbf{u}\|_2^2 \right\}, \tag{3.3}$$

where $\mathbb{1}_{\boldsymbol{\Theta}}(\mathbf{u})$ is the set indicator function on $\boldsymbol{\Theta}$, i.e., $\mathbb{1}_{\boldsymbol{\Theta}}(\mathbf{u}) = 0$ if $\mathbf{u} \in \boldsymbol{\Theta}$ and $\mathbb{1}_{\boldsymbol{\Theta}}(\mathbf{u}) = +\infty$ otherwise. $\eta > 0$ is any finite real value and is chosen as the step size in our paper. It is easy to see that $\mathbb{1}_{\boldsymbol{\Theta}}(\cdot)$ is nonsmooth. At the end of the $s$-th epoch, we update the reference policy as $\widetilde{\boldsymbol{\theta}}^{s+1} = \boldsymbol{\theta}_m^{s+1}$, where $\boldsymbol{\theta}_m^{s+1}$ is the last iterate of this epoch.

The goal of our algorithm is to find a point $\boldsymbol{\theta} \in \boldsymbol{\Theta}$ that maximizes the performance function $J(\boldsymbol{\theta})$ subject to the constraint, namely, $\max_{\boldsymbol{\theta} \in \boldsymbol{\Theta}} J(\boldsymbol{\theta}) = \max_{\boldsymbol{\theta} \in \mathbb{R}^d} \{J(\boldsymbol{\theta}) - \mathbb{1}_{\boldsymbol{\Theta}}(\boldsymbol{\theta})\}$. The gradient norm $\|\nabla J(\boldsymbol{\theta})\|_2$ is not sufficient to characterize the convergence of the algorithm due to additional the constraint. Following the literature on nonsmooth optimization (Reddi et al., 2016b; Ghadimi et al., 2016; Nguyen et al., 2017; Li & Li, 2018; Wang et al., 2018), we use the generalized first-order stationary condition: $\mathcal{G}_\eta(\boldsymbol{\theta}) = \mathbf{0}$, where the *gradient mapping* $\mathcal{G}_\eta$ is defined as follows

$$\mathcal{G}_\eta(\boldsymbol{\theta}) = \frac{1}{\eta}(\mathcal{P}_{\boldsymbol{\Theta}}(\boldsymbol{\theta} + \eta \nabla J(\boldsymbol{\theta})) - \boldsymbol{\theta}). \tag{3.4}$$

We can view $\mathcal{G}_\eta$ as a generalized projected gradient at $\boldsymbol{\theta}$. By definition if $\boldsymbol{\Theta} = \mathbb{R}^d$, we have $\mathcal{G}_\eta(\boldsymbol{\theta}) \equiv \nabla J(\boldsymbol{\theta})$. Therefore, the policy is update is displayed in Line 10 in Algorithm 1, where

---

**Algorithm 1** Stochastic Recursive Variance Reduced Policy Gradient (SRVR-PG)

---

1: **Input:** number of epochs $S$, epoch size $m$, step size $\eta$, batch size $N$, mini-batch size $B$, gradient estimator $g$, initial parameter $\widetilde{\boldsymbol{\theta}}^0 = \boldsymbol{\theta}_0 \in \boldsymbol{\Theta}$
2: **for** $s = 0, \ldots, S - 1$ **do**
3: $\quad \boldsymbol{\theta}_0^{s+1} = \widetilde{\boldsymbol{\theta}}^s$
4: $\quad$ Sample $N$ trajectories $\{\tau_i\}$ from $p(\cdot|\widetilde{\boldsymbol{\theta}}^s)$
5: $\quad \mathbf{v}_0^{s+1} = \widehat{\nabla}_{\boldsymbol{\theta}} J(\widetilde{\boldsymbol{\theta}}^s) := 1/N \sum_{i=1}^N g(\tau_i|\widetilde{\boldsymbol{\theta}}^s)$
6: $\quad \boldsymbol{\theta}_1^{s+1} = \mathcal{P}_{\boldsymbol{\Theta}}(\boldsymbol{\theta}_0^{s+1} + \eta \mathbf{v}_0^{s+1})$
7: $\quad$ **for** $t = 1, \ldots, m - 1$ **do**
8: $\quad\quad$ Sample $B$ trajectories $\{\tau_j\}$ from $p(\cdot|\boldsymbol{\theta}_t^{s+1})$
9: $\quad\quad \mathbf{v}_t^{s+1} = \mathbf{v}_{t-1}^{s+1} + \frac{1}{B} \sum_{j=1}^B \left( g(\tau_j|\boldsymbol{\theta}_t^{s+1}) - g_\omega(\tau_j|\boldsymbol{\theta}_{t-1}^{s+1}) \right)$
10: $\quad\quad \boldsymbol{\theta}_{t+1}^{s+1} = \mathcal{P}_{\boldsymbol{\Theta}}(\boldsymbol{\theta}_t^{s+1} + \eta \mathbf{v}_t^{s+1})$
11: $\quad$ **end for**
12: $\quad \widetilde{\boldsymbol{\theta}}^{s+1} = \boldsymbol{\theta}_m^{s+1}$
13: **end for**
14: **return** $\boldsymbol{\theta}_{\text{out}}$, which is uniformly picked from $\{\boldsymbol{\theta}_t^s\}_{t=0,\ldots,m-1;s=0,\ldots,S}$

---

prox is the proximal operator defined in (3.3). Similar recursive semi-stochastic gradients to (3.1) were first proposed in stochastic optimization for finite-sum problems, leading to the stochastic recursive gradient algorithm (SARAH) (Nguyen et al., 2017; 2019) and the stochastic path-integrated differential estimator (SPIDER) (Fang et al., 2018; Wang et al., 2018). However, our gradient estimator in (3.1) is noticeably different from that in Nguyen et al. (2017); Fang et al. (2018); Wang et al. (2018); Nguyen et al. (2019) due to the gradient estimator $g_\omega(\tau_j|\boldsymbol{\theta}_{t-1}^{s+1})$ defined in (3.2) that is equipped with step-wise importance weights. This term is essential to deal with the non-stationarity of the distribution of the trajectory $\tau$. Specifically, $\{\tau_j\}_{j=1}^B$ are sampled from policy $\pi_{\boldsymbol{\theta}_t^{s+1}}$ while the PGT/GPOMDP estimator $g(\cdot|\boldsymbol{\theta}_{t-1}^{s+1})$ is defined based on policy $\pi_{\boldsymbol{\theta}_{t-1}^{s+1}}$ according to (2.6). This inconsistency introduces extra challenges in the convergence analysis of SRVR-PG. Using importance weighting, we can obtain

$$\mathbb{E}_{\tau \sim p(\tau|\boldsymbol{\theta}_t^{s+1})}[g_\omega(\tau|\boldsymbol{\theta}_{t-1}^{s+1})] = \mathbb{E}_{\tau \sim p(\tau|\boldsymbol{\theta}_{t-1}^{s+1})}[g(\tau|\boldsymbol{\theta}_{t-1}^{s+1})],$$

which eliminates the inconsistency caused by the varying trajectory distribution.

It is worth noting that the semi-stochastic gradient in (3.1) also differs from the one used in SVRPG (Papini et al., 2018) because we recursively update $\mathbf{v}_t^{s+1}$ using $\mathbf{v}_{t-1}^{s+1}$ from the previous iteration, while SVRPG uses a reference gradient that is only updated at the beginning of each epoch. Moreover, SVRPG wastes $N$ trajectories without updating the policy at the beginning of each epoch, while Algorithm 1 updates the policy immediately after this sampling process (Line 6), which saves computation in practice.

We notice that very recently another algorithm called SARAPO (Yuan et al., 2019) is proposed which also uses a recursive gradient update in trust region policy optimization (Schulman et al., 2015a). Our Algorithm 1 differs from their algorithm at least in the following ways: (1) our recursive gradient $\mathbf{v}_t^s$ defined in (3.1) has an importance weight from the snapshot gradient while SARAPO does not; (2) we are optimizing the expected return while Yuan et al. (2019) optimizes the total advantage over state visitation distribution and actions under KullbackLeibler divergence constraint; and most importantly (3) there is no convergence or sample complexity analysis for SARAPO.

## 4 MAIN THEORY

In this section, we present the theoretical analysis of Algorithm 1. We first introduce some common assumptions used in the convergence analysis of policy gradient methods.

**Assumption 4.1.** Let $\pi_{\boldsymbol{\theta}}(a|s)$ be the policy parameterized by $\boldsymbol{\theta}$. There exist constants $G, M > 0$ such that the gradient and Hessian matrix of $\log \pi_{\boldsymbol{\theta}}(a|s)$ with respect to $\boldsymbol{\theta}$ satisfy

$$\|\nabla_{\boldsymbol{\theta}} \log \pi_{\boldsymbol{\theta}}(a|s)\| \leq G, \qquad \|\nabla_{\boldsymbol{\theta}}^2 \log \pi_{\boldsymbol{\theta}}(a|s)\|_2 \leq M,$$

for all $a \in \mathcal{A}$ and $s \in \mathcal{S}$.

The above boundedness assumption is reasonable since we usually require the policy function to be twice differentiable and easy to optimize in practice. Similarly, in Papini et al. (2018), the authors assume that $\frac{\partial}{\partial \theta_i} \log \pi_{\boldsymbol{\theta}}(a|s)$ and $\frac{\partial^2}{\partial \theta_i \partial \theta_j} \log \pi_{\boldsymbol{\theta}}(a|s)$ are upper bounded elementwisely, which is actually stronger than our Assumption 4.1.

In the following proposition, we show that Assumption 4.1 directly implies that the Hessian matrix of the performance function $\nabla^2 J(\boldsymbol{\theta})$ is bounded, which is often referred to as the smoothness assumption and is crucial in analyzing the convergence of nonconvex optimization (Reddi et al., 2016a; Allen-Zhu & Hazan, 2016).

**Proposition 4.2.** Let $g(\tau|\boldsymbol{\theta})$ be the PGT estimator defined in (2.5). Assumption 4.1 implies:

(1). $\|g(\tau|\boldsymbol{\theta}_1) - g(\tau|\boldsymbol{\theta}_2)\|_2 \leq L\|\boldsymbol{\theta}_1 - \boldsymbol{\theta}_2\|_2, \forall \boldsymbol{\theta}_1, \boldsymbol{\theta}_2 \in \mathbb{R}^d$, where the smoothness parameter is $L = MR/(1-\gamma)^2 + 2G^2 R/(1-\gamma)^3$;

(2). $J(\boldsymbol{\theta})$ is $L$-smooth, namely $\|\nabla^2_{\boldsymbol{\theta}} J(\boldsymbol{\theta})\|_2 \leq L$;

(3). $\|g(\tau|\boldsymbol{\theta})\|_2 \leq C_g$ for all $\boldsymbol{\theta} \in \mathbb{R}^d$, with $C_g = GR/(1-\gamma)^2$.

Similar properties are also proved in Xu et al. (2019). However, in contrast to their results, the smoothness parameter $L$ and the bound on the gradient norm here do not rely on horizon $H$. When $H \approx 1/(1-\gamma)$ and $\gamma$ is sufficiently close to 1, we can see that the order of the smoothness parameter is $O(1/(1-\gamma)^3)$, which matches the order $O(H^2/(1-\gamma))$ in Xu et al. (2019). The next assumption requires the variance of the gradient estimator is bounded.

**Assumption 4.3.** There exists a constant $\xi > 0$ such that $\text{Var}\big(g(\tau|\boldsymbol{\theta})\big) \leq \xi^2$, for all policy $\pi_{\boldsymbol{\theta}}$.

In Algorithm 1, we have used importance sampling to connect the trajectories between two different iterations. The following assumption ensures that the variance of the importance weight is bounded, which is also made in Papini et al. (2018); Xu et al. (2019).

**Assumption 4.4.** Let $\omega(\cdot|\boldsymbol{\theta}_1, \boldsymbol{\theta}_2) = p(\cdot|\boldsymbol{\theta}_1)/p(\cdot|\boldsymbol{\theta}_2)$. There is a constant $W < \infty$ such that for each policy pairs encountered in Algorithm 1,

$$\text{Var}(\omega(\tau|\boldsymbol{\theta}_1, \boldsymbol{\theta}_2)) \leq W, \quad \forall \boldsymbol{\theta}_1, \boldsymbol{\theta}_2 \in \mathbb{R}^d, \tau \sim p(\cdot|\boldsymbol{\theta}_2).$$

### 4.1 CONVERGENCE RATE AND SAMPLE COMPLEXITY OF SRVR-PG

Now we are ready to present the convergence result of SRVR-PG to a stationary point:

**Theorem 4.5.** Suppose that Assumptions 4.1, 4.3 and 4.4 hold. In Algorithm 1, we choose the step size $\eta \leq 1/(4L)$ and epoch size $m$ and mini-batch size $B$ such that

$$B \geq \frac{72m\eta G^2(2G^2/M + 1)(W+1)\gamma}{(1-\gamma)^2}.$$

Then the generalized projected gradient of the output of Algorithm 1 satisfies

$$\mathbb{E}\big[\big\|\mathcal{G}_\eta\big(\boldsymbol{\theta}_{\text{out}}\big)\big\|_2^2\big] \leq \frac{8[J(\boldsymbol{\theta}^*) - J(\boldsymbol{\theta}_0) - \mathbb{1}_{\boldsymbol{\Theta}}(\boldsymbol{\theta}^*) + \mathbb{1}_{\boldsymbol{\Theta}}(\boldsymbol{\theta}_0)]}{\eta S m} + \frac{6\xi^2}{N},$$

where $\boldsymbol{\theta}^* = \text{argmax}_{\boldsymbol{\theta} \in \boldsymbol{\Theta}} J(\boldsymbol{\theta})$.

**Remark 4.6.** Theorem 4.5 states that under a proper choice of step size, batch size and epoch length, the expected squared gradient norm of the performance function at the output of SRVR-PG is in the order of

$$O\left(\frac{1}{Sm} + \frac{1}{N}\right).$$

Recall that $S$ is the number of epochs and $m$ is the epoch length of SRVR-PG, so $Sm$ is the total number of iterations of SRVR-PG. Thus the first term $O(1/(Sm))$ characterizes the convergence rate of SRVR-PG. The second term $O(1/N)$ comes from the variance of the stochastic gradient used in the outer loop, where $N$ is the batch size used in the snapshot gradient $\mathbf{v}_0^{s+1}$ in Line 5 of SRVR-PG. Compared with the $O(1/(Sm) + 1/N + 1/B)$ convergence rate in Papini et al. (2018),

our analysis avoids the additional term $O(1/B)$ that depends on the mini-batch size within each epoch.

Compared with Xu et al. (2019), our mini-batch size $B$ is independent of the horizon length $H$. This enables us to choose a smaller mini-batch size $B$ while maintaining the same convergence rate. As we will show in the next corollary, this improvement leads to a lower sample complexity.

**Corollary 4.7.** Suppose the same conditions as in Theorem 4.5 hold. Set step size as $\eta = 1/(4L)$, the batch size parameters as $N = O(1/\epsilon)$ and $B = O(1/\epsilon^{1/2})$ respectively, epoch length as $m = O(1/\epsilon^{1/2})$ and the number of epochs as $S = O(1/\epsilon^{1/2})$. Then Algorithm 1 outputs a point $\boldsymbol{\theta}_{\text{out}}$ that satisfies $\mathbb{E}[\|\mathcal{G}_\eta(\boldsymbol{\theta}_{\text{out}})\|_2^2] \leq \epsilon$ within $O(1/\epsilon^{3/2})$ trajectories in total.

Note that the results in Papini et al. (2018); Xu et al. (2019) are for $\|\nabla_{\boldsymbol{\theta}} J(\boldsymbol{\theta})\|_2^2 \leq \epsilon$, while our result in Corollary 4.7 is more general. In particular, when the policy parameter $\boldsymbol{\theta}$ is defined on the whole space $\mathbb{R}^d$ instead of $\boldsymbol{\Theta}$, our result reduces to the case for $\|\nabla_{\boldsymbol{\theta}} J(\boldsymbol{\theta})\|_2^2 \leq \epsilon$ since $\boldsymbol{\Theta} = \mathbb{R}^d$ and $\mathcal{G}_\eta(\boldsymbol{\theta}) = \nabla_{\boldsymbol{\theta}} J(\boldsymbol{\theta})$. In Xu et al. (2019), the authors improved the sample complexity of SVRPG (Papini et al., 2018) from $O(1/\epsilon^2)$ to $O(1/\epsilon^{5/3})$ by a sharper analysis. According to Corollary 4.7, SRVR-PG only needs $O(1/\epsilon^{3/2})$ number of trajectories to achieve $\|\nabla_{\boldsymbol{\theta}} J(\boldsymbol{\theta})\|_2^2 \leq \epsilon$, which is lower than the sample complexity of SVRPG by a factor of $O(1/\epsilon^{1/6})$. This improvement is more pronounced when the required precision $\epsilon$ is very small.

## 4.2 IMPLICATION FOR GAUSSIAN POLICY

Now, we consider the Gaussian policy model and present the sample complexity of SRVR-PG in this setting. For bounded action space $\mathcal{A} \subset \mathbb{R}$, a Gaussian policy parameterized by $\boldsymbol{\theta}$ is defined as

$$\pi_{\boldsymbol{\theta}}(a|s) = \frac{1}{\sqrt{2\pi}} \exp\left(-\frac{(\boldsymbol{\theta}^\top \phi(s) - a)^2}{2\sigma^2}\right), \tag{4.1}$$

where $\sigma^2$ is a fixed standard deviation parameter and $\phi : \mathcal{S} \mapsto \mathbb{R}^d$ is a mapping from the state space to the feature space. For Gaussian policy, under the mild condition that the actions and the state feature vectors are bounded, we can verify that Assumptions 4.1 and 4.3 hold, which can be found in Appendix D. It is worth noting that Assumption 4.4 does not hold trivially for all Gaussian distributions. In particular, Cortes et al. (2010) showed that for two Gaussian distributions $\pi_{\boldsymbol{\theta}_1}(a|s) \sim N(\mu_1, \sigma_1^2)$ and $\pi_{\boldsymbol{\theta}_2}(a|s) \sim N(\mu_2, \sigma_2^2)$, if $\sigma_2 > \sqrt{2}/2\sigma_1$, then the variance of $\omega(\tau|\boldsymbol{\theta}_1, \boldsymbol{\theta}_2)$ is bounded. For our Gaussian policy defined in (4.1) where the standard deviation $\sigma^2$ is fixed, we have $\sigma > \sqrt{2}/2\sigma$ trivially hold, and therefore Assumption 4.4 holds for some finite constant $W > 0$ according to (2.1).

Recall that Theorem 4.5 holds for any general models under Assumptions 4.1, 4.3 and 4.4. Based on the above arguments, we know that the convergence analysis in Theorem 4.5 applies to Gaussian policy. In the following corollary, we present the sample complexity of Algorithm 1 for Gaussian policy with detailed dependency on precision parameter $\epsilon$, horizon size $H$ and the discount factor $\gamma$.

**Corollary 4.8.** Given the Gaussian policy defined in (4.1), suppose Assumption 4.4 holds and we have $|a| \leq C_a$ for all $a \in \mathcal{A}$ and $\|\phi(s)\|_2 \leq M_\phi$ for all $s \in \mathcal{S}$, where $C_a, M_\phi > 0$ are constants. If we set step size as $\eta = O((1-\gamma)^3)$, the mini-batch sizes and epoch length as $N = O((1-\gamma)^{-3}\epsilon^{-1})$, $B = O((1-\gamma)^{-1}\epsilon^{-1/2})$ and $m = O((1-\gamma)^{-2}\epsilon^{-1/2})$, then the output of Algorithm 1 satisfies $\mathbb{E}[\|\mathcal{G}_\eta(\boldsymbol{\theta}_{\text{out}})\|_2^2] \leq \epsilon$ after $O(1/((1-\gamma)^4\epsilon^{3/2}))$ trajectories in total.

**Remark 4.9.** For Gaussian policy, the number of trajectories Algorithm 1 needs to find an $\epsilon$-approximate stationary point, i.e., $\mathbb{E}[\|\mathcal{G}_\eta(\boldsymbol{\theta}_{\text{out}})\|_2^2] \leq \epsilon$, is also in the order of $O(\epsilon^{-3/2})$, which is faster than PGT and SVRPG. Additionally, we explicitly show that the sample complexity does not depend on the horizon $H$, which is in sharp contrast with the results in Papini et al. (2018); Xu et al. (2019). The dependence on $1/(1-\gamma)$ comes from the variance of PGT estimator.

## 5 EXPERIMENTS

In this section, we provide experiment results of the proposed algorithm on benchmark reinforcement learning environments including the Cartpole, Mountain Car and Pendulum problems. In all

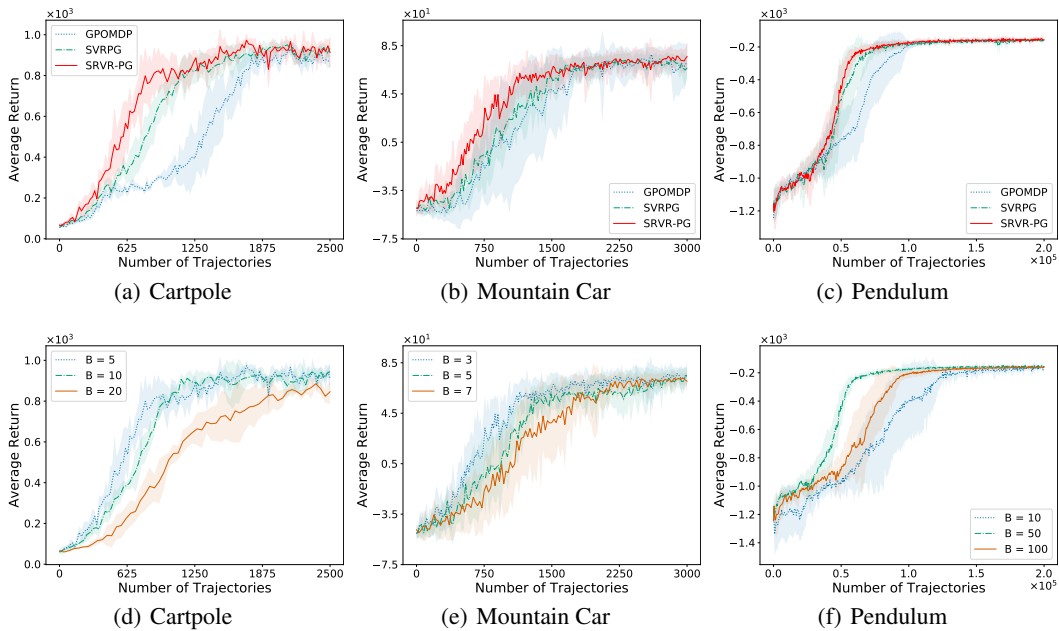

Figure 1: (a)-(c): Comparison of different algorithms. Experimental results are averaged over 10 repetitions. (d)-(f): Comparison of different batch size $B$ on the performance of SRVR-PG.

the experiments, we use the Gaussian policy defined in (4.1). In addition, we found that the proposed algorithm works well without the extra projection step. Therefore, we did not use projection in our experiments. For baselines, we compare the proposed SRVR-PG algorithm with the most relevant methods: GPOMDP (Baxter & Bartlett, 2001) and SVRPG (Papini et al., 2018). For the learning rates $\eta$ in all of our experiments, we use grid search to directly tune $\eta$. For instance, we searched $\eta$ for the Cartpole problem by evenly dividing the interval $[10^{-5}, 10^{-1}]$ into 20 points in the log-space. For the batch size parameters $N$ and $B$ and the epoch length $m$, according to Corollary 4.7, we choose $N = O(1/\epsilon)$, $B = O(1/\epsilon^{1/2})$ and thus $m = O(1/\epsilon^{1/2})$, where $\epsilon > 0$ is a user-defined precision parameter. In our experiments, we set $N = C_0/\epsilon$, $B = C_1/\epsilon^{1/2}$ and $m = C_2/\epsilon^{1/2}$ and tune the constant parameters $C_0, C_1, C_2$ using grid search. The detailed parameters used in the experiments are presented in Appendix E.

We evaluate the performance of different algorithms in terms of the total number of trajectories they require to achieve a certain threshold of cumulative rewards. We run each experiment repeatedly for 10 times and plot the averaged returns with standard deviation. For a given environment, all experiments are initialized from the same random initialization. Figures 1(a), 1(b) and 1(c) show the results on the comparison of GPOMDP, SVRPG, and our proposed SRVR-PG algorithm across three different RL environments. It is evident that, for all environments, GPOMDP is overshadowed by the variance reduced algorithms SVRPG and SRVR-PG significantly. Furthermore, SRVR-PG outperforms SVRPG in all experiments, which is consistent with the comparison on the sample complexity of GPOMDP, SVRPG and SRVR-PG in Table 1.

Corollaries 4.7 and 4.8 suggest that when the mini-batch size $B$ is in the order of $O(\sqrt{N})$, SRVR-PG achieves the best performance. Here $N$ is the number of episodes sampled in the outer loop of Algorithm 1 and $B$ is the number of episodes sampled at each inner loop iteration. To validate our theoretical result, we conduct a sensitivity study to demonstrate the effectiveness of different batch sizes within each epoch of SRVR-PG on its performance. The results on different environments are displayed in Figures 1(d), 1(e) and 1(f) respectively. To interpret these results, we take the Pendulum problem as an example. In this setting, we choose outer loop batch size $N$ of Algorithm 1 to be $N = 250$. By Corollary 4.8, the optimal choice of batch size in the inner loop of Algorithm 1 is $B = C\sqrt{N}$, where $C > 1$ is a constant depending on horizon $H$ and discount factor $\gamma$. Figure 1(f)

shows that $B = 50 \approx 3\sqrt{N}$ yields the best convergence results for SRVR-PG on Pendulum, which validates our theoretical analysis and implies that a larger batch size $B$ does not necessarily result in an improvement in sample complexity, as each update requires more trajectories, but a smaller batch size $B$ pushes SRVR-PG to behave more similar to GPOMDP. Moreover, by comparing with the outer loop batch size $N$ presented in Table 2 for SRVR-PG in Cartpole and Mountain Car environments, we found that the results in Figures 1(d) and 1(e) are again in alignment with our theory. Due to the space limit, additional experiment results are included in Appendix E.

## 6  CONCLUSIONS

We propose a novel policy gradient method called SRVR-PG, which is built on a recursively updated stochastic policy gradient estimator. We prove that the sample complexity of SRVR-PG is lower than the sample complexity of the state-of-the-art SVRPG (Papini et al., 2018; Xu et al., 2019) algorithm. We also extend the new variance reduction technique to policy gradient with parameter-based exploration and propose the SRVR-PG-PE algorithm, which outperforms the original PGPE algorithm both in theory and practice. Experiments on the classic reinforcement learning benchmarks validate the advantage of our proposed algorithms.

### ACKNOWLEDGMENTS

We would like to thank the anonymous reviewers for their helpful comments. We would also like to thank Rui Yuan for pointing out an error on the calculation of the smoothness parameter for the performance function in the previous version. This research was sponsored in part by the National Science Foundation IIS-1904183, IIS-1906169 and Adobe Data Science Research Award. The views and conclusions contained in this paper are those of the authors and should not be interpreted as representing any funding agencies.

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

## A    EXTENSION TO PARAMETER-BASED EXPLORATION

Although SRVR-PG is proposed for action-based policy gradient, it can be easily extended to the policy gradient algorithm with parameter-based exploration (PGPE) (Sehnke et al., 2008). Unlike action-based policy gradient in previous sections, PGPE does not directly optimize the policy parameter $\boldsymbol{\theta}$ but instead assumes that it follows a prior distribution with hyper-parameter $\boldsymbol{\rho}$: $\boldsymbol{\theta} \sim p(\boldsymbol{\theta}|\boldsymbol{\rho})$. The expected return under the policy induced by the hyper-parameter $\boldsymbol{\rho}$ is formulated as follows[2]

$$J(\boldsymbol{\rho}) = \int \int p(\boldsymbol{\theta}|\boldsymbol{\rho})p(\tau|\boldsymbol{\theta})\mathcal{R}(\tau)\mathrm{d}\tau\mathrm{d}\boldsymbol{\theta}. \tag{A.1}$$

PGPE aims to find the hyper-parameter $\boldsymbol{\rho}^*$ that maximizes the performance function $J(\boldsymbol{\rho})$. Since $p(\boldsymbol{\theta}|\boldsymbol{\rho})$ is stochastic and can provide sufficient exploration, we can choose $\pi_{\boldsymbol{\theta}}(a|s) = \delta(a - \mu_{\boldsymbol{\theta}}(s))$ to be a deterministic policy, where $\delta$ is the Dirac delta function and $\mu_{\boldsymbol{\theta}}(\cdot)$ is a deterministic function. For instance, a linear deterministic policy is defined as $\pi_{\boldsymbol{\theta}}(a|s) = \delta(a - \boldsymbol{\theta}^\top s)$ (Zhao et al., 2011; Metelli et al., 2018). Given the policy parameter $\boldsymbol{\theta}$, a trajectory $\tau$ is only decided by the initial state distribution and the transition probability. Therefore, PGPE is called a parameter-based exploration approach. Similar to the action-based policy gradient methods, we can apply gradient ascent to find $\boldsymbol{\rho}^*$. In the $k$-th iteration, we update $\boldsymbol{\rho}_k$ by $\boldsymbol{\rho}_{k+1} = \boldsymbol{\rho}_k + \eta \nabla_{\boldsymbol{\rho}} J(\boldsymbol{\rho})$. The exact gradient of $J(\boldsymbol{\rho})$ with respect to $\boldsymbol{\rho}$ is given by

$$\nabla_{\boldsymbol{\rho}} J(\boldsymbol{\rho}) = \int \int p(\boldsymbol{\theta}|\boldsymbol{\rho})p(\tau|\boldsymbol{\theta})\nabla_{\boldsymbol{\rho}} \log p(\boldsymbol{\theta}|\boldsymbol{\rho})\mathcal{R}(\tau)\mathrm{d}\tau\mathrm{d}\boldsymbol{\theta}.$$

To approximate $\nabla_{\boldsymbol{\rho}} J(\boldsymbol{\rho})$, we first sample $N$ policy parameters $\{\boldsymbol{\theta}_i\}$ from $p(\boldsymbol{\theta}|\boldsymbol{\rho})$. Then we sample one trajectory $\tau_i$ for each $\boldsymbol{\theta}_i$ and use the following empirical average to approximate $\nabla_{\boldsymbol{\rho}} J(\boldsymbol{\rho})$

$$\widehat{\nabla}_{\boldsymbol{\rho}} J(\boldsymbol{\rho}) = \frac{1}{N} \sum_{i=1}^{N} \nabla_{\boldsymbol{\rho}} \log p(\boldsymbol{\theta}_i|\boldsymbol{\rho}) \sum_{h=0}^{H} \gamma^h r(s_h^i, a_h^i) := \frac{1}{N} \sum_{i=1}^{N} g(\tau_i|\boldsymbol{\rho}), \tag{A.2}$$

where $\gamma \in [0, 1)$ is the discount factor. Compared with the PGT/GPOMDP estimator in Section 2, the likelihood term $\nabla_{\boldsymbol{\rho}} \log p(\boldsymbol{\theta}_i|\boldsymbol{\rho})$ in (A.2) for PGPE is independent of horizon $H$.

Algorithm 1 can be directly applied to the PGPE setting, where we replace the policy parameter $\boldsymbol{\theta}$ with the hyper-parameter $\boldsymbol{\rho}$. When we need to sample $N$ trajectories, we first sample $N$ policy parameters $\{\boldsymbol{\theta}_i\}$ from $p(\boldsymbol{\theta}|\boldsymbol{\rho})$. Since the policy is deterministic with given $\boldsymbol{\theta}_i$, we sample one trajectory $\tau_i$ from each policy $p(\tau|\boldsymbol{\theta}_i)$. The recursive semi-stochastic gradient is given by

$$\mathbf{v}_t^{s+1} = \frac{1}{B} \sum_{j=1}^{B} g(\tau_j|\boldsymbol{\rho}_t^{s+1}) - \frac{1}{B} \sum_{j=1}^{B} g_\omega(\tau_j|\boldsymbol{\rho}_{t-1}^{s+1}) + \mathbf{v}_{t-1}^{s+1}, \tag{A.3}$$

where $g_\omega(\tau_j|\boldsymbol{\rho}_{t-1}^{s+1})$ is the gradient estimator with step-wise importance weight defined in the way as in (3.2). We call this variance reduced parameter-based algorithm SRVR-PG-PE, which is displayed in Algorithm 2.

Under similar assumptions on the parameter distribution $p(\boldsymbol{\theta}|\boldsymbol{\rho})$, as Assumptions 4.1, 4.3 and 4.4, we can easily prove that SRVR-PG-PE converges to a stationary point of $J(\boldsymbol{\rho})$ with $O(1/\epsilon^{3/2})$ sample complexity. In particular, we assume the policy parameter $\boldsymbol{\theta}$ follows the distribution $p(\boldsymbol{\theta}|\boldsymbol{\rho})$ and we update our estimation of $\boldsymbol{\rho}$ based on the semi-stochastic gradient in (A.3). Recall the gradient $\widehat{\nabla}_{\boldsymbol{\rho}} J(\boldsymbol{\rho})$ derived in (A.2). Since the policy in SRVR-PG-PE is deterministic, we only need to make the boundedness assumption on $p(\boldsymbol{\theta}|\boldsymbol{\rho})$. In particular, we assume that

1. $\|\nabla_{\boldsymbol{\rho}} \log p(\boldsymbol{\theta}|\boldsymbol{\rho})\|_2$ and $\|\nabla_{\boldsymbol{\rho}}^2 \log p(\boldsymbol{\theta}|\boldsymbol{\rho})\|_2$ are bounded by constants in a similar way to Assumption 4.1;

2. the gradient estimator $g(\tau|\boldsymbol{\rho}) = \nabla_{\boldsymbol{\rho}} \log p(\boldsymbol{\theta}|\boldsymbol{\rho}) \sum_{h=0}^{H} \gamma^h r(s_h, a_h)$ has bounded variance;

3. and the importance weight $\omega(\tau_j|\boldsymbol{\rho}_{t-1}^{s+1}, \boldsymbol{\rho}_t^{s+1}) = p(\boldsymbol{\theta}_j|\boldsymbol{\rho}_{t-1}^{s+1})/p(\boldsymbol{\theta}_j|\boldsymbol{\rho}_t^{s+1})$ has bounded variance in a similar way to Assumption 4.4.

---

[2]We slightly abuse the notation by overloading $J$ as the performance function defined on the hyper-parameter $\boldsymbol{\rho}$.

Then the same gradient complexity $O(1/\epsilon^{3/2})$ for SRVR-PG-PE can be proved in the same way as the proof of Theorem 4.5 and Corollary 4.7. Since the analysis is almost the same as that of SRVR-PG, we omit the proof of the convergence of SRVR-PG-PE. In fact, according to the analysis in Zhao et al. (2011); Metelli et al. (2018), all the three assumptions listed above can be easily verified under a Gaussian prior for $\boldsymbol{\theta}$ and a linear deterministic policy.

---

**Algorithm 2** Stochastic Recursive Variance Reduced Policy Gradient with Parameter-based Exploration (SRVR-PG-PE)

---

1: **Input:** number of epochs $S$, epoch size $m$, step size $\eta$, batch size $N$, mini-batch size $B$, gradient estimator $g$, initial parameter $\boldsymbol{\rho}_m^0 := \widetilde{\boldsymbol{\rho}}^0 := \boldsymbol{\rho}_0$
2: **for** $s = 0, \ldots, S-1$ **do**
3:     $\boldsymbol{\rho}_0^{s+1} = \boldsymbol{\rho}^s$
4:     Sample $N$ policy parameters $\{\boldsymbol{\theta}_i\}$ from $p(\cdot|\boldsymbol{\rho}^s)$
5:     Sample one trajectory $\tau_i$ from each policy $\pi_{\boldsymbol{\theta}_i}$
6:     $\mathbf{v}_0^{s+1} = \widehat{\nabla}_{\boldsymbol{\rho}} J(\boldsymbol{\rho}^s) := \frac{1}{N} \sum_{i=1}^{N} g(\tau_i|\widetilde{\boldsymbol{\rho}}^s)$
7:     $\boldsymbol{\rho}_1^{s+1} = \boldsymbol{\rho}_0^{s+1} + \eta \mathbf{v}_0^{s+1}$
8:     **for** $t = 1, \ldots, m-1$ **do**
9:         Sample $B$ policy parameters $\{\boldsymbol{\theta}_j\}$ from $p(\cdot|\boldsymbol{\rho}_t^{s+1})$
10:        Sample one trajectory $\tau_j$ from each policy $\pi_{\boldsymbol{\theta}_j}$
11:        $\mathbf{v}_t^{s+1} = \mathbf{v}_{t-1}^{s+1} + \frac{1}{B} \sum_{j=1}^{B} \left( g(\tau_j|\boldsymbol{\rho}_t^{s+1}) - g_\omega(\tau_j|\boldsymbol{\rho}_{t-1}^{s+1}) \right)$
12:        $\boldsymbol{\rho}_{t+1}^{s+1} = \boldsymbol{\rho}_t^{s+1} + \eta \mathbf{v}_t^{s+1}$
13:     **end for**
14: **end for**
15: **return** $\boldsymbol{\rho}_{\text{out}}$, which is uniformly picked from $\{\boldsymbol{\rho}_t^s\}_{t=0,\ldots,m;s=0,\ldots,S}$

---

# B    PROOF OF THE MAIN THEORY

In this section, we provide the proofs of the theoretical results for SRVR-PG (Algorithm 1). Before we start the proof of Theorem 4.5, we first lay down the following key lemma that controls the variance of the importance sampling weight $\omega$.

**Lemma B.1.** For any $\boldsymbol{\theta}_1, \boldsymbol{\theta}_2 \in \mathbb{R}^d$, let $\omega_{0:h}(\tau|\boldsymbol{\theta}_1, \boldsymbol{\theta}_2) = p(\tau_h|\boldsymbol{\theta}_1)/p(\tau_h|\boldsymbol{\theta}_2)$, where $\tau_h$ is a truncated trajectory of $\tau$ up to step $h$. Under Assumptions 4.1 and 4.4, it holds that

$$\mathrm{Var}\big(\omega_{0:h}(\tau|\boldsymbol{\theta}_1, \boldsymbol{\theta}_2)\big) \le C_\omega \|\boldsymbol{\theta}_1 - \boldsymbol{\theta}_2\|_2^2,$$

where $C_\omega = h(2hG^2 + M)(W+1)$.

Recall that in Assumption 4.4 we assume the variance of the importance weight is upper bounded by a constant $W$. Based on this assumption, Lemma B.1 further bounds the variance of the importance weight via the distance between the behavioral and the target policies. As the algorithm converges, these two policies will be very close and the bound in Lemma B.1 could be much tighter than the constant bound.

*Proof of Theorem 4.5.* By plugging the definition of the projection operator in (3.3) into the update rule $\boldsymbol{\theta}_{t+1}^{s+1} = \mathcal{P}_\Theta\big(\boldsymbol{\theta}_t^{s+1} + \eta \mathbf{v}_t^{s+1}\big)$, we have

$$\boldsymbol{\theta}_{t+1}^{s+1} = \underset{\mathbf{u} \in \mathbb{R}^d}{\operatorname{argmin}} \, \mathbb{1}_\Theta(\mathbf{u}) + 1/(2\eta)\big\|\mathbf{u} - \boldsymbol{\theta}_t^{s+1}\big\|_2^2 - \langle \mathbf{v}_t^{s+1}, \mathbf{u} \rangle. \tag{B.1}$$

Similar to the generalized projected gradient $\mathcal{G}_\eta(\boldsymbol{\theta})$ defined in (3.4), we define $\widetilde{\mathcal{G}}_t^{s+1}$ to be a (stochastic) gradient mapping based on the recursive gradient estimator $\mathbf{v}_t^{s+1}$:

$$\widetilde{\mathcal{G}}_t^{s+1} = \frac{1}{\eta}\big(\boldsymbol{\theta}_{t+1}^{s+1} - \boldsymbol{\theta}_t^{s+1}\big) = \frac{1}{\eta}\big(\mathcal{P}_\Theta\big(\boldsymbol{\theta}_t^{s+1} + \eta \mathbf{v}_t^{s+1}\big) - \boldsymbol{\theta}_t^{s+1}\big). \tag{B.2}$$

The definition of $\widetilde{\mathcal{G}}_t^{s+1}$ differs from $\mathcal{G}_\eta(\boldsymbol{\theta}_t^{s+1})$ only in the semi-stochastic gradient term $\mathbf{v}_t^{s+1}$, while the latter one uses the full gradient $\nabla J(\boldsymbol{\theta}_t^{s+1})$. Note that $\mathbb{1}_\Theta(\cdot)$ is convex but not smooth. We

assume that $\mathbf{p} \in \partial \mathbb{1}_\Theta(\boldsymbol{\theta}_{t+1}^{s+1})$ is a sub-gradient of $\mathbb{1}_\Theta(\cdot)$. According to the optimality condition of (B.1), we have $\mathbf{p} + 1/\eta(\boldsymbol{\theta}_{t+1}^{s+1} - \boldsymbol{\theta}_t^{s+1}) - \mathbf{v}_t^{s+1} = \mathbf{0}$. Further by the convexity of $\mathbb{1}_\Theta(\cdot)$, we have

$$
\begin{aligned}
\mathbb{1}_\Theta(\boldsymbol{\theta}_{t+1}^{s+1}) &\leq \mathbb{1}_\Theta(\boldsymbol{\theta}_t^{s+1}) + \langle \mathbf{p}, \boldsymbol{\theta}_{t+1}^{s+1} - \boldsymbol{\theta}_t^{s+1} \rangle \\
&= \mathbb{1}_\Theta(\boldsymbol{\theta}_t^{s+1}) - \langle 1/\eta(\boldsymbol{\theta}_{t+1}^{s+1} - \boldsymbol{\theta}_t^{s+1}) - \mathbf{v}_t^{s+1}, \boldsymbol{\theta}_{t+1}^{s+1} - \boldsymbol{\theta}_t^{s+1} \rangle. \quad (B.3)
\end{aligned}
$$

By Proposition 4.2, $J(\boldsymbol{\theta})$ is $L$-smooth, which by definition directly implies

$$
J(\boldsymbol{\theta}_{t+1}^{s+1}) \geq J(\boldsymbol{\theta}_t^{s+1}) + \langle \nabla J(\boldsymbol{\theta}_t^{s+1}), \boldsymbol{\theta}_{t+1}^{s+1} - \boldsymbol{\theta}_t^{s+1} \rangle - \frac{L}{2} \|\boldsymbol{\theta}_{t+1}^{s+1} - \boldsymbol{\theta}_t^{s+1}\|_2^2.
$$

For the simplification of presentation, let us define the notation $\Phi(\boldsymbol{\theta}) = J(\boldsymbol{\theta}) - \mathbb{1}_\Theta(\boldsymbol{\theta})$. Then according to the definition of $\mathbb{1}_\Theta$ we have $\operatorname{argmax}_{\boldsymbol{\theta} \in \mathbb{R}^d} \Phi(\boldsymbol{\theta}) = \operatorname{argmax}_{\boldsymbol{\theta} \in \Theta} J(\boldsymbol{\theta}) := \boldsymbol{\theta}^*$. Combining the above inequality with (B.3), we have

$$
\begin{aligned}
\Phi(\boldsymbol{\theta}_{t+1}^{s+1}) &\geq \Phi(\boldsymbol{\theta}_t^{s+1}) + \langle \nabla J(\boldsymbol{\theta}_t^{s+1}) - \mathbf{v}_t^{s+1}, \boldsymbol{\theta}_{t+1}^{s+1} - \boldsymbol{\theta}_t^{s+1} \rangle + \left(\frac{1}{\eta} - \frac{L}{2}\right)\|\boldsymbol{\theta}_{t+1}^{s+1} - \boldsymbol{\theta}_t^{s+1}\|_2^2 \\
&= \Phi(\boldsymbol{\theta}_t^{s+1}) + \langle \nabla J(\boldsymbol{\theta}_t^{s+1}) - \mathbf{v}_t^{s+1}, \eta \widetilde{\mathcal{G}}_t^{s+1} \rangle + \eta \|\widetilde{\mathcal{G}}_t^{s+1}\|_2^2 - \frac{L}{2}\|\boldsymbol{\theta}_{t+1}^{s+1} - \boldsymbol{\theta}_t^{s+1}\|_2^2 \\
&\geq \Phi(\boldsymbol{\theta}_t^{s+1}) - \frac{\eta}{2}\|\nabla J(\boldsymbol{\theta}_t^{s+1}) - \mathbf{v}_t^{s+1}\|_2^2 + \frac{\eta}{2}\|\widetilde{\mathcal{G}}_t^{s+1}\|_2^2 - \frac{L}{2}\|\boldsymbol{\theta}_{t+1}^{s+1} - \boldsymbol{\theta}_t^{s+1}\|_2^2 \\
&= \Phi(\boldsymbol{\theta}_t^{s+1}) - \frac{\eta}{2}\|\nabla J(\boldsymbol{\theta}_t^{s+1}) - \mathbf{v}_t^{s+1}\|_2^2 + \frac{\eta}{4}\|\widetilde{\mathcal{G}}_t^{s+1}\|_2^2 + \left(\frac{1}{4\eta} - \frac{L}{2}\right)\|\boldsymbol{\theta}_{t+1}^{s+1} - \boldsymbol{\theta}_t^{s+1}\|_2^2 \\
&\geq \Phi(\boldsymbol{\theta}_t^{s+1}) - \frac{\eta}{2}\|\nabla J(\boldsymbol{\theta}_t^{s+1}) - \mathbf{v}_t^{s+1}\|_2^2 + \frac{\eta}{8}\|\mathcal{G}_\eta(\boldsymbol{\theta}_t^{s+1})\|_2^2 \\
&\quad - \frac{\eta}{4}\|\mathcal{G}_\eta(\boldsymbol{\theta}_t^{s+1}) - \widetilde{\mathcal{G}}_t^{s+1}\|_2^2 + \left(\frac{1}{4\eta} - \frac{L}{2}\right)\|\boldsymbol{\theta}_{t+1}^{s+1} - \boldsymbol{\theta}_t^{s+1}\|_2^2, \quad (B.4)
\end{aligned}
$$

where the second inequality holds due to Young's inequality and the third inequality holds due to the fact that $\|\mathcal{G}_\eta(\boldsymbol{\theta}_t^{s+1})\|_2^2 \leq 2\|\widetilde{\mathcal{G}}_t^{s+1}\|_2^2 + 2\|\mathcal{G}_\eta(\boldsymbol{\theta}_t^{s+1}) - \widetilde{\mathcal{G}}_t^{s+1}\|_2^2$. Denote $\bar{\boldsymbol{\theta}}_{t+1}^{s+1} = \operatorname{prox}_{\eta \mathbb{1}_\Theta}(\boldsymbol{\theta}_t^{s+1} + \eta \nabla J(\boldsymbol{\theta}_t^{s+1}))$. By similar argument in (B.3) we have

$$
\begin{aligned}
\mathbb{1}_\Theta(\boldsymbol{\theta}_{t+1}^{s+1}) &\leq \mathbb{1}_\Theta(\bar{\boldsymbol{\theta}}_{t+1}^{s+1}) - \langle 1/\eta(\boldsymbol{\theta}_{t+1}^{s+1} - \boldsymbol{\theta}_t^{s+1}) - \mathbf{v}_t^{s+1}, \boldsymbol{\theta}_{t+1}^{s+1} - \bar{\boldsymbol{\theta}}_{t+1}^{s+1} \rangle, \\
\mathbb{1}_\Theta(\bar{\boldsymbol{\theta}}_{t+1}^{s+1}) &\leq \mathbb{1}_\Theta(\boldsymbol{\theta}_{t+1}^{s+1}) - \langle 1/\eta(\boldsymbol{\theta}_{t+1}^{s+1} - \boldsymbol{\theta}_t^{s+1}) - \nabla J(\boldsymbol{\theta}_t^{s+1}), \bar{\boldsymbol{\theta}}_{t+1}^{s+1} - \boldsymbol{\theta}_{t+1}^{s+1} \rangle.
\end{aligned}
$$

Adding the above two inequalities immediately yields $\|\bar{\boldsymbol{\theta}}_{t+1}^{s+1} - \boldsymbol{\theta}_{t+1}^{s+1}\|_2 \leq \eta\|\nabla J(\boldsymbol{\theta}_t^{s+1}) - \mathbf{v}_t^{s+1}\|_2$, which further implies $\|\mathcal{G}_\eta(\boldsymbol{\theta}_t^{s+1}) - \widetilde{\mathcal{G}}_t^{s+1}\|_2 \leq \|\nabla J(\boldsymbol{\theta}_t^{s+1}) - \mathbf{v}_t^{s+1}\|_2$. Submitting this result into (B.4), we obtain

$$
\begin{aligned}
\Phi(\boldsymbol{\theta}_{t+1}^{s+1}) &\geq \Phi(\boldsymbol{\theta}_t^{s+1}) - \frac{3\eta}{4}\|\nabla J(\boldsymbol{\theta}_t^{s+1}) - \mathbf{v}_t^{s+1}\|_2^2 + \frac{\eta}{8}\|\mathcal{G}_\eta(\boldsymbol{\theta}_t^{s+1})\|_2^2 \\
&\quad + \left(\frac{1}{4\eta} - \frac{L}{2}\right)\|\boldsymbol{\theta}_{t+1}^{s+1} - \boldsymbol{\theta}_t^{s+1}\|_2^2. \quad (B.5)
\end{aligned}
$$

We denote the index set of $\{\tau_j\}_{j=1}^B$ in the $t$-th inner iteration by $\mathcal{B}_t$. Note that

$$
\begin{aligned}
&\|\nabla J(\boldsymbol{\theta}_t^{s+1}) - \mathbf{v}_t^{s+1}\|_2^2 \\
&= \left\|\nabla J(\boldsymbol{\theta}_t^{s+1}) - \mathbf{v}_{t-1}^{s+1} + \frac{1}{B}\sum_{j \in \mathcal{B}_t}\left(g_\omega(\tau_j|\boldsymbol{\theta}_{t-1}^{s+1}) - g(\tau_j|\boldsymbol{\theta}_t^{s+1})\right)\right\|_2^2 \\
&= \left\|\nabla J(\boldsymbol{\theta}_t^{s+1}) - \nabla J(\boldsymbol{\theta}_{t-1}^{s+1}) + \frac{1}{B}\sum_{j \in \mathcal{B}_t}\left(g_\omega(\tau_j|\boldsymbol{\theta}_{t-1}^{s+1}) - g(\tau_j|\boldsymbol{\theta}_t^{s+1})\right) + \nabla J(\boldsymbol{\theta}_{t-1}^{s+1}) - \mathbf{v}_{t-1}^{s+1}\right\|_2^2 \\
&= \left\|\nabla J(\boldsymbol{\theta}_t^{s+1}) - \nabla J(\boldsymbol{\theta}_{t-1}^{s+1}) + \frac{1}{B}\sum_{j \in \mathcal{B}_t}\left(g_\omega(\tau_j|\boldsymbol{\theta}_{t-1}^{s+1}) - g(\tau_j|\boldsymbol{\theta}_t^{s+1})\right)\right\|^2 \\
&\quad + \frac{2}{B}\sum_{j \in \mathcal{B}_t}\left\langle \nabla J(\boldsymbol{\theta}_t^{s+1}) - \nabla J(\boldsymbol{\theta}_{t-1}^{s+1}) + g_\omega(\tau_j|\boldsymbol{\theta}_{t-1}^{s+1}) - g(\tau_j|\boldsymbol{\theta}_t^{s+1}), \nabla J(\boldsymbol{\theta}_{t-1}^{s+1}) - \mathbf{v}_{t-1}^{s+1} \right\rangle
\end{aligned}
$$

$$+ \left\| \nabla J(\boldsymbol{\theta}_{t-1}^{s+1}) - \mathbf{v}_{t-1}^{s+1} \right\|_2^2. \tag{B.6}$$

Conditional on $\boldsymbol{\theta}_t^{s+1}$, taking the expectation over $\mathcal{B}_t$ yields

$$\mathbb{E}\big[ \big\langle \nabla J(\boldsymbol{\theta}_t^{s+1}) - g(\tau_j | \boldsymbol{\theta}_t^{s+1}), \nabla J(\boldsymbol{\theta}_{t-1}^{s+1}) - \mathbf{v}_{t-1}^{s+1} \big\rangle \big] = 0.$$

Similarly, taking the expectation over $\boldsymbol{\theta}_t^{s+1}$ and the choice of $\mathcal{B}_t$ yields

$$\mathbb{E}\big[ \big\langle \nabla J(\boldsymbol{\theta}_{t-1}^{s+1}) - g_\omega(\tau_j | \boldsymbol{\theta}_{t-1}^{s+1}), \nabla J(\boldsymbol{\theta}_{t-1}^{s+1}) - \mathbf{v}_{t-1}^{s+1} \big\rangle \big] = 0.$$

Combining the above equations with (B.6), we obtain

$$\mathbb{E}\big[ \big\| \nabla J(\boldsymbol{\theta}_t^{s+1}) - \mathbf{v}_t^{s+1} \big\|_2^2 \big]$$

$$= \mathbb{E}\left\| \nabla J(\boldsymbol{\theta}_t^{s+1}) - \nabla J(\boldsymbol{\theta}_{t-1}^{s+1}) + \frac{1}{B} \sum_{j \in \mathcal{B}_t} \big( g_\omega(\tau_j | \boldsymbol{\theta}_{t-1}^{s+1}) - g(\tau_j | \boldsymbol{\theta}_t^{s+1}) \big) \right\|_2^2$$

$$+ \mathbb{E}\big\| \nabla J(\boldsymbol{\theta}_{t-1}^{s+1}) - \mathbf{v}_{t-1}^{s+1} \big\|_2^2$$

$$= \frac{1}{B^2} \sum_{j \in \mathcal{B}_t} \mathbb{E}\big\| \nabla J(\boldsymbol{\theta}_t^{s+1}) - \nabla J(\boldsymbol{\theta}_{t-1}^{s+1}) + g_\omega(\tau_j | \boldsymbol{\theta}_{t-1}^{s+1}) - g(\tau_j | \boldsymbol{\theta}_t^{s+1}) \big\|_2^2$$

$$+ \mathbb{E}\big\| \nabla J(\boldsymbol{\theta}_{t-1}^{s+1}) - \mathbf{v}_{t-1}^{s+1} \big\|_2^2, \tag{B.7}$$

$$\leq \frac{1}{B^2} \sum_{j \in \mathcal{B}_t} \mathbb{E}\big\| g_\omega(\tau_j | \boldsymbol{\theta}_{t-1}^{s+1}) - g(\tau_j | \boldsymbol{\theta}_t^{s+1}) \big\|_2^2 + \big\| \nabla J(\boldsymbol{\theta}_{t-1}^{s+1}) - \mathbf{v}_{t-1}^{s+1} \big\|_2^2, \tag{B.8}$$

where (B.7) is due to the fact that $\mathbb{E}\|\mathbf{x}_1 + \ldots + \mathbf{x}_n\|_2^2 = \mathbb{E}\|\mathbf{x}_1\|_2 + \ldots + \mathbb{E}\|\mathbf{x}_n\|_2$ for independent zero-mean random variables, and (B.8) holds due to the fact that $\mathbf{x}_1, \ldots, \mathbf{x}_n$ is due to $\mathbb{E}\|\mathbf{x} - \mathbb{E}\mathbf{x}\|_2^2 \leq \mathbb{E}\|\mathbf{x}\|_2^2$. For the first term, we have $\big\| g_\omega(\tau_j | \boldsymbol{\theta}_{t-1}^{s+1}) - g(\tau_j | \boldsymbol{\theta}_t^{s+1}) \big\|_2 \leq \big\| g_\omega(\tau_j | \boldsymbol{\theta}_{t-1}^{s+1}) - g(\tau_j | \boldsymbol{\theta}_{t-1}^{s+1}) \big\|_2 + L\big\| \boldsymbol{\theta}_{t-1}^{s+1} - \boldsymbol{\theta}_t^{s+1} \big\|_2$ by triangle inequality and Proposition 4.2.

$$\mathbb{E}\big[ \big\| g_\omega(\tau_j | \boldsymbol{\theta}_{t-1}^{s+1}) - g(\tau_j | \boldsymbol{\theta}_{t-1}^{s+1}) \big\|_2^2 \big] = \mathbb{E}\left[ \left\| \sum_{h=0}^{H-1} (\omega_{0:h} - 1) \left[ \sum_{t=0}^{h} \nabla_{\boldsymbol{\theta}} \log \pi_{\boldsymbol{\theta}}(a_t^i | s_t^i) \right] \gamma^h r(s_h^i, a_h^i) \right\|_2^2 \right]$$

$$= \sum_{h=0}^{H-1} \mathbb{E}\left[ \left\| (\omega_{0:h} - 1) \left[ \sum_{t=0}^{h} \nabla_{\boldsymbol{\theta}} \log \pi_{\boldsymbol{\theta}}(a_t^i | s_t^i) \right] \gamma^h r(s_h^i, a_h^i) \right\|_2^2 \right]$$

$$\leq \sum_{h=0}^{H-1} h^2 (2G^2 + M)(W + 1) \big\| \boldsymbol{\theta}_{t-1}^{s+1} - \boldsymbol{\theta}_t^{s+1} \big\|_2^2 \cdot h^2 G^2 \gamma^h R$$

$$\leq \frac{24 R G^2 (2G^2 + M)(W + 1)\gamma}{(1 - \gamma)^5} \big\| \boldsymbol{\theta}_{t-1}^{s+1} - \boldsymbol{\theta}_t^{s+1} \big\|_2^2, \tag{B.9}$$

where in the second equality we used the fact that $\mathbb{E}[\nabla \log \pi_{\boldsymbol{\theta}}(a|s)] = \mathbf{0}$, the first inequality is due to Lemma B.1 and in the last inequality we use the fact that $\sum_{h=0}^{\infty} h^4 \gamma^h = \gamma(\gamma^3 + 11\gamma^2 + 11\gamma + 1)/(1 - \gamma)^5$ for $|\gamma| < 1$. Combining the results in (B.8) and (B.9), we get

$$\mathbb{E}\big\| \nabla J(\boldsymbol{\theta}_t^{s+1}) - \mathbf{v}_t^{s+1} \big\|_2^2 \leq \frac{C_\gamma}{B} \big\| \boldsymbol{\theta}_t^{s+1} - \boldsymbol{\theta}_{t-1}^{s+1} \big\|_2^2 + \big\| \nabla J(\boldsymbol{\theta}_{t-1}^{s+1}) - \mathbf{v}_{t-1}^{s+1} \big\|_2^2$$

$$\leq \frac{C_\gamma}{B} \sum_{l=1}^{t} \big\| \boldsymbol{\theta}_l^{s+1} - \boldsymbol{\theta}_{l-1}^{s+1} \big\|_2^2 + \big\| \nabla J(\boldsymbol{\theta}_0^{s+1}) - \mathbf{v}_0^{s+1} \big\|_2^2, \tag{B.10}$$

which holds for $t = 1, \ldots, m - 1$, where $C_\gamma = 24 R G^2 (2G^2 + M)(W + 1)\gamma/(1 - \gamma)^5$. According to Algorithm 1 and Assumption 4.3, we have

$$\mathbb{E}\big\| \nabla J(\boldsymbol{\theta}_0^{s+1}) - \mathbf{v}_0^{s+1} \big\|_2^2 \leq \frac{\xi^2}{N}. \tag{B.11}$$

Submitting the above result into (B.5) yields

$$\mathbb{E}_{N,B}\big[ \Phi(\boldsymbol{\theta}_{t+1}^{s+1}) \big] \geq \mathbb{E}_{N,B}\big[ \Phi(\boldsymbol{\theta}_t^{s+1}) \big] + \frac{\eta}{8} \big\| \mathcal{G}_\eta(\boldsymbol{\theta}_t^{s+1}) \big\|_2^2 + \left( \frac{1}{4\eta} - \frac{L}{2} \right) \big\| \boldsymbol{\theta}_{t+1}^{s+1} - \boldsymbol{\theta}_t^{s+1} \big\|_2^2$$

$$-\frac{3\eta C_\gamma}{4B}\mathbb{E}_{N,B}\left[\sum_{l=1}^{t}\big\|\boldsymbol{\theta}_l^{s+1}-\boldsymbol{\theta}_{l-1}^{s+1}\big\|_2^2\right]-\frac{3\eta\xi^2}{4N}, \tag{B.12}$$

for $t = 1, \ldots, m - 1$. Recall Line 6 in Algorithm 1, where we update $\boldsymbol{\theta}_1^{t+1}$ with the average of a mini-batch of gradients $\mathbf{v}_0^s = 1/N\sum_{i=1}^{N}g(\tau_i|\widetilde{\boldsymbol{\theta}}^s)$. Similar to (B.5), by smoothness of $J(\boldsymbol{\theta})$, we have

$$\Phi(\boldsymbol{\theta}_1^{s+1}) \geq \Phi(\boldsymbol{\theta}_0^{s+1}) - \frac{3\eta}{4}\big\|\nabla J(\boldsymbol{\theta}_0^{s+1}) - \mathbf{v}_0^{s+1}\big\|_2^2 + \frac{\eta}{8}\big\|\mathcal{G}_\eta(\boldsymbol{\theta}_0^{s+1})\big\|_2^2$$
$$+ \left(\frac{1}{4\eta} - \frac{L}{2}\right)\big\|\boldsymbol{\theta}_1^{s+1} - \boldsymbol{\theta}_0^{s+1}\big\|_2^2.$$

Further by (B.11), it holds that

$$\mathbb{E}\big[\Phi(\boldsymbol{\theta}_1^{s+1})\big] \geq \mathbb{E}\big[\Phi(\boldsymbol{\theta}_0^{s+1})\big] - \frac{3\eta\xi^2}{4N} + \frac{\eta}{8}\big\|\mathcal{G}_\eta(\boldsymbol{\theta}_0^{s+1})\big\|_2^2 + \left(\frac{1}{4\eta} - \frac{L}{2}\right)\big\|\boldsymbol{\theta}_1^{s+1} - \boldsymbol{\theta}_0^{s+1}\big\|_2^2. \tag{B.13}$$

Telescoping inequality (B.12) from $t = 1$ to $m - 1$ and combining the result with (B.13), we obtain

$$\mathbb{E}_{N,B}\big[\Phi(\boldsymbol{\theta}_m^{s+1})\big] \geq \mathbb{E}_{N,B}\big[\Phi(\boldsymbol{\theta}_0^{s+1})\big] + \frac{\eta}{8}\sum_{t=0}^{m-1}\mathbb{E}_N\big[\big\|\mathcal{G}_\eta(\boldsymbol{\theta}_t^{s+1})\big\|_2^2\big] - \frac{3m\eta\xi^2}{4N}$$
$$+ \left(\frac{1}{4\eta} - \frac{L}{2}\right)\sum_{t=0}^{m-1}\big\|\boldsymbol{\theta}_{t+1}^{s+1} - \boldsymbol{\theta}_t^{s+1}\big\|_2^2$$
$$- \frac{3\eta C_\gamma}{2B}\mathbb{E}_{N,B}\left[\sum_{t=0}^{m-1}\sum_{l=1}^{t}\big\|\boldsymbol{\theta}_l^{s+1} - \boldsymbol{\theta}_{l-1}^{s+1}\big\|_2^2\right]$$
$$\geq \mathbb{E}_{N,B}\big[\Phi(\boldsymbol{\theta}_0^{s+1})\big] + \frac{\eta}{8}\sum_{t=0}^{m-1}\mathbb{E}_N\big[\big\|\mathcal{G}_\eta(\boldsymbol{\theta}_t^{s+1})\big\|_2^2\big] - \frac{3m\eta\xi^2}{4N}$$
$$+ \left(\frac{1}{4\eta} - \frac{L}{2} - \frac{3m\eta C_\gamma}{2B}\right)\sum_{t=0}^{m-1}\big\|\boldsymbol{\theta}_{t+1}^{s+1} - \boldsymbol{\theta}_t^{s+1}\big\|_2^2. \tag{B.14}$$

If we choose step size $\eta$ and the epoch length $B$ such that

$$\eta \leq \frac{1}{4L}, \qquad \frac{B}{m} \geq \frac{3\eta C_\gamma}{L} = \frac{72\eta G^2(2G^2 + M)(W + 1)\gamma}{M(1 - \gamma)^2}, \tag{B.15}$$

and note that $\boldsymbol{\theta}_0^{s+1} = \widetilde{\boldsymbol{\theta}}^s$, $\boldsymbol{\theta}_m^{s+1} = \widetilde{\boldsymbol{\theta}}^{s+1}$, then (B.14) leads to

$$\mathbb{E}_N\big[\Phi(\widetilde{\boldsymbol{\theta}}^{s+1})\big] \geq \mathbb{E}_N\big[\Phi(\widetilde{\boldsymbol{\theta}}^s)\big] + \frac{\eta}{8}\sum_{t=0}^{m-1}\mathbb{E}_N\big[\big\|\mathcal{G}_\eta(\boldsymbol{\theta}_t^{s+1})\big\|_2^2\big] - \frac{3m\eta\xi^2}{4N}. \tag{B.16}$$

Summing up the above inequality over $s = 0, \ldots, S - 1$ yields

$$\frac{\eta}{8}\sum_{s=0}^{S-1}\sum_{t=0}^{m-1}\mathbb{E}\big[\big\|\mathcal{G}_\eta(\boldsymbol{\theta}_t^{s+1})\big\|_2^2\big] \leq \mathbb{E}\big[\Phi(\widetilde{\boldsymbol{\theta}}^S)\big] - \mathbb{E}\big[\Phi(\widetilde{\boldsymbol{\theta}}^0)\big] + \frac{3Sm\eta\xi^2}{4N},$$

which immediately implies

$$\mathbb{E}\big[\big\|\mathcal{G}_\eta(\boldsymbol{\theta}_{\text{out}})\big\|_2^2\big] \leq \frac{8\big(\mathbb{E}\big[\Phi(\widetilde{\boldsymbol{\theta}}^S)\big] - \mathbb{E}\big[\Phi(\widetilde{\boldsymbol{\theta}}^0)\big]\big)}{\eta Sm} + \frac{6\xi^2}{N} \leq \frac{8(\Phi(\boldsymbol{\theta}^*) - \Phi(\boldsymbol{\theta}_0))}{\eta Sm} + \frac{6\xi^2}{N}.$$

This completes the proof. $\square$

*Proof of Corollary 4.7.* Based on the convergence results in Theorem 4.5, in order to ensure $\mathbb{E}\big[\big\|\nabla J(\boldsymbol{\theta}_{\text{out}})\big\|_2^2\big] \leq \epsilon$, we can choose $S, m$ and $N$ such that

$$\frac{8(J(\boldsymbol{\theta}^*) - J(\boldsymbol{\theta}_0))}{\eta Sm} = \frac{\epsilon}{2}, \qquad \frac{6\xi^2}{N} = \frac{\epsilon}{2},$$

which implies $Sm = O(1/\epsilon)$ and $N = O(1/\epsilon)$. Note that we have set $m = O(B)$. The total number of stochastic gradient evaluations $\mathcal{T}_g$ we need is

$$\mathcal{T}_g = SN + SmB = O\left(\frac{N}{B\epsilon} + \frac{B}{\epsilon}\right) = O\left(\frac{1}{\epsilon^{3/2}}\right),$$

where we set $B = 1/\epsilon^{1/2}$. $\qquad\qquad\qquad\qquad\qquad\qquad\qquad\qquad\qquad\qquad\qquad\square$

## C    PROOF OF TECHNICAL LEMMAS

In this section, we provide the proofs of the technical lemmas. We first prove the smoothness of the performance function $J(\boldsymbol{\theta})$.

*Proof of Proposition 4.2.* Recall the definition of PGT in (2.5). We first show the Lipschitzness of $g(\tau|\boldsymbol{\theta})$ with baseline $b = 0$ as follows:

$$\begin{aligned}
\|\nabla g(\tau|\boldsymbol{\theta})\|_2 &= \left\| \sum_{h=0}^{H-1} \nabla_{\boldsymbol{\theta}}^2 \log \pi_{\boldsymbol{\theta}}(a_h|s_h) \left( \sum_{t=h}^{H-1} \gamma^t r(s_t, a_t) \right) \right\|_2 \\
&\leq \left( \sum_{t=0}^{H-1} \gamma^h \|\nabla_{\boldsymbol{\theta}}^2 \log \pi_{\boldsymbol{\theta}}(a_t|s_t)\|_2 \right) \frac{R}{1-\gamma} \\
&\leq \frac{MR}{(1-\gamma)^2},
\end{aligned}$$

where we used the fact that $0 < \gamma < 1$. When we have a nonzero baseline $b_h$, we can simply scale it with $\gamma^h$ and the above result still holds up to a constant multiplier.

Since the PGT estimator is an unbiased estimator of the policy gradient $\nabla_{\boldsymbol{\theta}} J(\boldsymbol{\theta})$, we have $\nabla_{\boldsymbol{\theta}} J(\boldsymbol{\theta}) = \mathbb{E}_\tau[g(\tau|\boldsymbol{\theta})]$ and thus

$$\begin{aligned}
\nabla_{\boldsymbol{\theta}}^2 J(\boldsymbol{\theta}) &= \int_\tau p(\tau|\boldsymbol{\theta}) \nabla_{\boldsymbol{\theta}} g(\tau|\boldsymbol{\theta}) \mathrm{d}\tau + \int_\tau p(\tau|\boldsymbol{\theta}) g(\tau|\boldsymbol{\theta}) \nabla_{\boldsymbol{\theta}} \log p(\tau|\boldsymbol{\theta}) \mathrm{d}\tau \\
&= \mathbb{E}_\tau[\nabla_{\boldsymbol{\theta}} g(\tau|\boldsymbol{\theta})] + \mathbb{E}_\tau[g(\tau|\boldsymbol{\theta}) \nabla_{\boldsymbol{\theta}} \log p(\tau|\boldsymbol{\theta})]. \qquad\qquad (\text{C.1})
\end{aligned}$$

We have already bounded the norm of the first term by $MR/(1-\gamma)^2$. Now we take a look at the second term. Plugging the equivalent definition of $g(\tau|\boldsymbol{\theta})$ in (2.6) yields

$$\begin{aligned}
&\mathbb{E}_\tau[g(\tau|\boldsymbol{\theta}) \nabla_{\boldsymbol{\theta}} \log p(\tau|\boldsymbol{\theta})] \\
&= \int_\tau \sum_{h=0}^{H-1} \left( \sum_{t=0}^{h} \nabla_{\boldsymbol{\theta}} \log \pi_{\boldsymbol{\theta}}(a_t|s_t) \right) \gamma^h r(s_h, a_h) \nabla_{\boldsymbol{\theta}} \log p(\tau|\boldsymbol{\theta}) \cdot p(\tau|\boldsymbol{\theta}) \mathrm{d}\tau \\
&= \int_\tau \sum_{h=0}^{H-1} \left( \sum_{t=0}^{h} \nabla_{\boldsymbol{\theta}} \log \pi_{\boldsymbol{\theta}}(a_t|s_t) \right) \gamma^h r(s_h, a_h) \sum_{t'=0}^{H-1} \nabla_{\boldsymbol{\theta}} \log \pi_{\boldsymbol{\theta}}(a_{t'}|s_{t'}) \cdot p(\tau|\boldsymbol{\theta}) \mathrm{d}\tau \\
&= \int_\tau \sum_{h=0}^{H-1} \left( \sum_{t=0}^{h} \nabla_{\boldsymbol{\theta}} \log \pi_{\boldsymbol{\theta}}(a_t|s_t) \right) \gamma^h r(s_h, a_h) \sum_{t'=0}^{h} \nabla_{\boldsymbol{\theta}} \log \pi_{\boldsymbol{\theta}}(a_{t'}|s_{t'}) \cdot p(\tau|\boldsymbol{\theta}) \mathrm{d}\tau, \qquad (\text{C.2})
\end{aligned}$$

where the second equality is due to $\nabla_{\boldsymbol{\theta}} P(s_{t'+1}|s_{t'}, a_{t'}) = 0$, and the last equality is due to the fact that for all $t' > h$ it holds that

$$\int_\tau \sum_{t=0}^{h} \nabla_{\boldsymbol{\theta}} \log \pi_{\boldsymbol{\theta}}(a_t|s_t) \gamma^h r(s_h, a_h) \nabla_{\boldsymbol{\theta}} \log \pi_{\boldsymbol{\theta}}(a_{t'}|s_{t'}) \cdot p(\tau|\boldsymbol{\theta}) \mathrm{d}\tau = 0.$$

Therefore, we have

$$\|\mathbb{E}_\tau[g(\tau|\boldsymbol{\theta}) \nabla_{\boldsymbol{\theta}} \log p(\tau|\boldsymbol{\theta})]\|_2 \leq \mathbb{E}_\tau\left[ \sum_{h=0}^{H-1} \sum_{t=0}^{h} G \gamma^h R \times (h+1) G \right]$$

$$= \sum_{h=0}^{H-1} G^2 R(h+1)^2 \gamma^h$$

$$\leq \frac{2G^2 R}{(1-\gamma)^3}.$$

Putting the above pieces together, we can obtain

$$\left\| \nabla_{\boldsymbol{\theta}}^2 J(\boldsymbol{\theta}) \right\|_2 \leq \frac{MR}{(1-\gamma)^2} + \frac{2G^2 R}{(1-\gamma)^3} := L,$$

which implies that $J(\boldsymbol{\theta})$ is $L$-smooth with $L = MR/(1-\gamma)^2 + 2G^2 R/(1-\gamma)^3$.

Similarly, we can bound the norm of gradient estimator as follows

$$\|g(\tau|\boldsymbol{\theta})\|_2 \leq \left\| \sum_{h=0}^{H-1} \nabla_{\boldsymbol{\theta}} \log \pi_{\boldsymbol{\theta}}(a_h|s_h) \frac{\gamma^h R(1-\gamma^{H-h})}{1-\gamma} \right\|_2 \leq \frac{GR}{(1-\gamma)^2},$$

which completes the proof. $\qquad\qquad\qquad\qquad\qquad\qquad\qquad\qquad\qquad\qquad\qquad\square$

**Lemma C.1** (Lemma 1 in Cortes et al. (2010)). Let $\omega(x) = P(x)/Q(x)$ be the importance weight for distributions $P$ and $Q$. Then $\mathbb{E}[\omega] = 1, \mathbb{E}[\omega^2] = d_2(P\|Q)$, where $d_2(P\|Q) = 2^{D_2(P\|Q)}$ and $D_2(P\|Q)$ is the Rényi divergence between distributions $P$ and $Q$. Note that this immediately implies $\mathrm{Var}(\omega) = d_2(P\|Q) - 1$.

*Proof of Lemma B.1.* According to the property of importance weight in Lemma C.1, we know

$$\mathrm{Var}\big(\omega_{0:h}\big(\tau|\widetilde{\boldsymbol{\theta}}^s, \boldsymbol{\theta}_t^{s+1}\big)\big) = d_2\big(p(\tau_h|\widetilde{\boldsymbol{\theta}}^s)\|p(\tau_h|\boldsymbol{\theta}_t^{s+1})\big) - 1.$$

To simplify the presentation, we denote $\boldsymbol{\theta}_1 = \widetilde{\boldsymbol{\theta}}^s$ and $\boldsymbol{\theta}_2 = \boldsymbol{\theta}_t^{s+1}$ in the rest of this proof. By definition, we have

$$d_2(p(\tau_h|\boldsymbol{\theta}_1)\|p(\tau_h|\boldsymbol{\theta}_2)) = \int_\tau p(\tau_h|\boldsymbol{\theta}_1) \frac{p(\tau_h|\boldsymbol{\theta}_1)}{p(\tau_h|\boldsymbol{\theta}_2)} \mathrm{d}\tau = \int_\tau p(\tau_h|\boldsymbol{\theta}_1)^2 p(\tau_h|\boldsymbol{\theta}_2)^{-1} \mathrm{d}\tau.$$

Taking the gradient of $d_2(p(\tau_h|\boldsymbol{\theta}_1)\|p(\tau_h|\boldsymbol{\theta}_2))$ with respect to $\boldsymbol{\theta}_1$, we have

$$\nabla_{\boldsymbol{\theta}_1} d_2(p(\tau_h|\boldsymbol{\theta}_1)\|p(\tau_h|\boldsymbol{\theta}_2)) = 2 \int_\tau p(\tau_h|\boldsymbol{\theta}_1) \nabla_{\boldsymbol{\theta}_1} p(\tau_h|\boldsymbol{\theta}_1) p(\tau_h|\boldsymbol{\theta}_2)^{-1} \mathrm{d}\tau.$$

In particular, if we set the value of $\boldsymbol{\theta}_1$ to be $\boldsymbol{\theta}_1 = \boldsymbol{\theta}_2$ in the above formula of the gradient, we get

$$\nabla_{\boldsymbol{\theta}_1} d_2(p(\tau_h|\boldsymbol{\theta}_1)\|p(\tau_h|\boldsymbol{\theta}_2))\big|_{\boldsymbol{\theta}_1 = \boldsymbol{\theta}_2} = 2 \int_\tau \nabla_{\boldsymbol{\theta}_1} p(\tau_h|\boldsymbol{\theta}_1) \mathrm{d}\tau\big|_{\boldsymbol{\theta}_1 = \boldsymbol{\theta}_2} = 0.$$

Applying mean value theorem with respect to the variable $\boldsymbol{\theta}_1$, we have

$$d_2(p(\tau_h|\boldsymbol{\theta}_1)\|p(\tau_h|\boldsymbol{\theta}_2)) = 1 + 1/2(\boldsymbol{\theta}_1 - \boldsymbol{\theta}_2)^\top \nabla_{\boldsymbol{\theta}}^2 d_2(p(\tau_h|\boldsymbol{\theta})\|p(\tau_h|\boldsymbol{\theta}_2))(\boldsymbol{\theta}_1 - \boldsymbol{\theta}_2), \qquad \text{(C.3)}$$

where $\boldsymbol{\theta} = t\boldsymbol{\theta}_1 + (1-t)\boldsymbol{\theta}_2$ for some $t \in [0, 1]$ and we used the fact that $d_2(p(\tau_h|\boldsymbol{\theta}_2)\|p(\tau_h|\boldsymbol{\theta}_2)) = 1$. To bound the above exponentiated Rényi divergence, we need to compute the Hessian matrix. Taking the derivative of $\nabla_{\boldsymbol{\theta}_1} d_2(p(\tau_h|\boldsymbol{\theta}_1)\|p(\tau_h|\boldsymbol{\theta}_2))$ with respect to $\boldsymbol{\theta}_1$ further yields

$$\nabla_{\boldsymbol{\theta}}^2 d_2(p(\tau_h|\boldsymbol{\theta})\|p(\tau_h|\boldsymbol{\theta}_2)) = 2 \int_\tau \nabla_{\boldsymbol{\theta}} \log p(\tau_h|\boldsymbol{\theta}) \nabla_{\boldsymbol{\theta}} \log p(\tau_h|\boldsymbol{\theta})^\top \frac{p(\tau_h|\boldsymbol{\theta})^2}{p(\tau_h|\boldsymbol{\theta}_2)} \mathrm{d}\tau$$

$$+ 2 \int_\tau \nabla_{\boldsymbol{\theta}}^2 p(\tau_h|\boldsymbol{\theta}) p(\tau_h|\boldsymbol{\theta}) p(\tau_h|\boldsymbol{\theta}_2)^{-1} \mathrm{d}\tau. \qquad \text{(C.4)}$$

Thus we need to compute the Hessian matrix of the trajectory distribution function, i.e., $\nabla_{\boldsymbol{\theta}}^2 p(\tau_h|\boldsymbol{\theta})$, which can further be derived from the Hessian matrix of the log-density function.

$$\nabla_{\boldsymbol{\theta}}^2 \log p(\tau_h|\boldsymbol{\theta}) = -p(\tau_h|\boldsymbol{\theta})^{-2} \nabla_{\boldsymbol{\theta}} p(\tau_h|\boldsymbol{\theta}) \nabla_{\boldsymbol{\theta}} p(\tau_h|\boldsymbol{\theta})^\top + p(\tau_h|\boldsymbol{\theta})^{-1} \nabla_{\boldsymbol{\theta}}^2 p(\tau_h|\boldsymbol{\theta}). \qquad \text{(C.5)}$$

Submitting (C.5) into (C.4) yields

$$
\begin{aligned}
\|\nabla_{\boldsymbol{\theta}}^2 d_2(p(\tau_h|\boldsymbol{\theta})\|p(\tau_h|\boldsymbol{\theta}_2))\|_2 &= \left\| 4\int_\tau \nabla_{\boldsymbol{\theta}} \log p(\tau_h|\boldsymbol{\theta}) \nabla_{\boldsymbol{\theta}} \log p(\tau_h|\boldsymbol{\theta})^\top \frac{p(\tau_h|\boldsymbol{\theta})^2}{p(\tau_h|\boldsymbol{\theta}_2)} \mathrm{d}\tau \right. \\
&\qquad \left. + 2\int_\tau \nabla_{\boldsymbol{\theta}}^2 \log p(\tau_h|\boldsymbol{\theta}) \frac{p(\tau_h|\boldsymbol{\theta})^2}{p(\tau_h|\boldsymbol{\theta}_2)} \mathrm{d}\tau \right\|_2 \\
&\leq \int_\tau \frac{p(\tau_h|\boldsymbol{\theta})^2}{p(\tau_h|\boldsymbol{\theta}_2)} \left( 4\|\nabla_{\boldsymbol{\theta}} \log p(\tau_h|\boldsymbol{\theta})\|_2^2 + 2\|\nabla_{\boldsymbol{\theta}}^2 \log p(\tau_h|\boldsymbol{\theta})\|_2 \right)\mathrm{d}\tau \\
&\leq (4h^2 G^2 + 2hM)\mathbb{E}[\omega(\tau|\boldsymbol{\theta},\boldsymbol{\theta}_2)^2] \\
&\leq 2h(2hG^2 + M)(W+1),
\end{aligned}
$$

where the second inequality comes from Assumption 4.1 and the last inequality is due to Assumption 4.4 and Lemma C.1. Combining the above result with (C.3), we have

$$
\mathrm{Var}\big(\omega_{0:h}\big(\tau|\widetilde{\boldsymbol{\theta}}^s,\boldsymbol{\theta}_t^{s+1}\big)\big) = d_2\big(p(\tau_h|\widetilde{\boldsymbol{\theta}}^s)\|p(\tau_h|\boldsymbol{\theta}_t^{s+1})\big) - 1 \leq C_\omega \|\widetilde{\boldsymbol{\theta}}^s - \boldsymbol{\theta}_t^{s+1}\|_2^2,
$$

where $C_\omega = h(2hG^2 + M)(W+1)$. $\qquad\square$

## D  PROOF OF THEORETICAL RESULTS FOR GAUSSIAN POLICY

In this section, we prove the sample complexity for Gaussian policy. According to (4.1), we can calculate the gradient and Hessian matrix of the logarithm of the policy.

$$
\nabla \log \pi_{\boldsymbol{\theta}}(a|s) = \frac{(a - \boldsymbol{\theta}^\top \phi(s))\phi(s)}{\sigma^2}, \quad \nabla^2 \log \pi_{\boldsymbol{\theta}}(a|s) = -\frac{\phi(s)\phi(s)^\top}{\sigma^2}. \tag{D.1}
$$

It is easy to see that Assumption 4.1 holds with $G = C_a M_\phi/\sigma^2$ and $M = M_\phi^2/\sigma^2$. Based on this observation, Proposition 4.2 also holds for Gaussian policy with parameters defined as follows

$$
L = \frac{RM_\phi^2}{\sigma^2(1-\gamma)^3}, \quad \text{and} \quad C_g = \frac{RC_a M_\phi}{\sigma^2(1-\gamma)^2}. \tag{D.2}
$$

The following lemma gives the variance $\xi^2$ of the PGT estimator, which verifies Assumption 4.3.

**Lemma D.1** (Lemma 5.5 in Pirotta et al. (2013)). Given a Gaussian policy $\pi_{\boldsymbol{\theta}}(a|s) \sim N(\boldsymbol{\theta}^\top \phi(s), \sigma^2)$, if the $|r(s,a)| \leq R$ and $\|\phi(s)\|_2 \leq M_\phi$ for all $s \in \mathcal{S}, a \in \mathcal{A}$ and $R, M_\phi > 0$ are constants, then the variance of PGT estimator defined in (2.5) can be bounded as follows:

$$
\mathrm{Var}(g(\tau|\boldsymbol{\theta})) \leq \xi^2 = \frac{R^2 M_\phi^2}{(1-\gamma)^2 \sigma^2} \left( \frac{1-\gamma^{2H}}{1-\gamma^2} - H\gamma^{2H} - 2\gamma^H \frac{1-\gamma^H}{1-\gamma} \right).
$$

*Proof of Corollary 4.8.* The proof will be similar to that of Corollary 4.7. By Theorem 4.5, to ensure that $\mathbb{E}[\|\nabla J(\boldsymbol{\theta}_{\mathrm{out}})\|_2^2] \leq \epsilon$, we can set

$$
\frac{8(J(\boldsymbol{\theta}^*) - J(\boldsymbol{\theta}_0))}{\eta Sm} = \frac{\epsilon}{2}, \quad \frac{6\xi^2}{N} = \frac{\epsilon}{2}.
$$

Plugging the value of $\xi^2$ in Lemma D.1 into the second equation above yields $N = O(\epsilon^{-1}(1-\gamma)^{-3})$. For the first equation, we have $S = O(1/(\eta m\epsilon))$. Therefore, the total number of stochastic gradient evaluations $\mathcal{T}_g$ required by Algorithm 1 is

$$
\mathcal{T}_g = SN + SmB = O\left( \frac{N}{\eta m\epsilon} + \frac{B}{\eta\epsilon} \right).
$$

So a good choice of batch size $B$ and epoch length $m$ will lead to $Bm = N$. Combining this with the requirement of $B$ in Theorem 4.5, we can set

$$
m = \sqrt{\frac{LN}{\eta C_\gamma}}, \quad \text{and} \quad B = \sqrt{\frac{N\eta C_\gamma}{L}}.
$$

Note that $C_\gamma = 24RG^2(2G^2 + M)(W + 1)\gamma/(1 - \gamma)^5$. Plugging the values of $G, N$ and L into the above equations yields

$$m = O\left(\frac{1}{(1 - \gamma)^2\sqrt{\epsilon}}\right), \quad B = O\left(\frac{1}{(1 - \gamma)^1\sqrt{\epsilon}}\right).$$

The corresponding sample complexity is

$$\mathcal{T}_g = O\left(\frac{1}{(1 - \gamma)^4\epsilon^{3/2}}\right).$$

This completes the proof for Gaussian policy. $\qquad\square$

## E  ADDITIONAL DETAILS ON EXPERIMENTS

Now, we provide more details of our experiments presented in Section 5. We first present the parameters for all algorithms we used in all our experiments in Tables 2 and 3. Among the parameters, the neural network structure and the RL environment parameters are shared across all the algorithms. As mentioned in Section 5, the order of the batch size parameters of our algorithm are chosen according to Corollary 4.7 and we multiply them by a tuning constant via grid search. Similarly, the orders of batch size parameters of SVRPG and GPOMDP are chosen based on the theoretical results suggested by Papini et al. (2018); Xu et al. (2019). Moerover, the learning rates for different methods are tuned by grid search.

We then present the results of PGPE and SRVR-PG-PE on Cartpole, Mountain Car and Pendulum in Figure 2. In all three environments, our SRVR-PG-PE algorithm shows improvement over PGPE (Sehnke et al., 2010) in terms of number of trajectories. It is worth noting that in all these environments both PGPE and SRVR-PG-PE seem to solve the problem very quickly, which is consistent with the results reported in (Zhao et al., 2011; 2013; Metelli et al., 2018). Our primary goal in this experiment is to show that our proposed variance reduced policy gradient algorithm can be easily extended to the PGPE framework. To avoid distracting the audience's attention from the variance reduction algorithm on the sample complexity, we do not thoroughly compare the performance of the parameter based policy gradient methods such as PGPE and SRVR-PG-PE with the action based policy gradient methods. We refer interested readers to the valuable empirical studies of PGPE based algorithms presented in Zhao et al. (2011; 2013); Metelli et al. (2018).

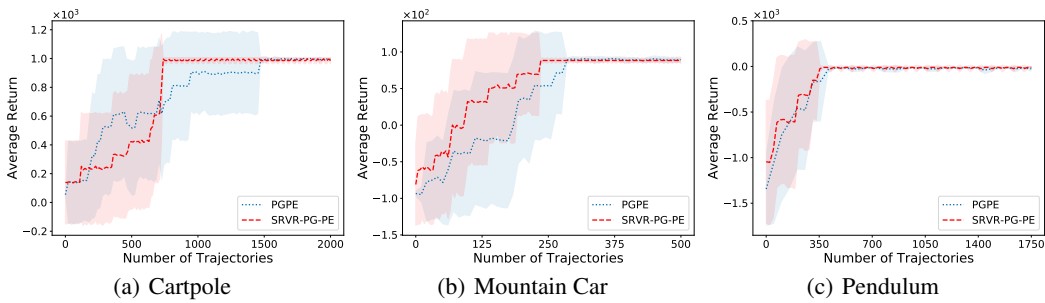

(a) Cartpole  (b) Mountain Car  (c) Pendulum

Figure 2: Performance of SRVR-PG-PE compared with PGPE. Experiment results are averaged over 10 runs.

Table 2: Parameters used in the SRVR-PG experiments.

| Parameters | Algorithm | Cartpole | Mountain Car | Pendulum |
|---|---|---|---|---|
| NN size | - | 64 | 64 | $8\times8$ |
| NN activation function | - | Tanh | Tanh | Tanh |
| Task horizon | - | 100 | 1000 | 200 |
| Total trajectories | - | 2500 | 3000 | $2 \times 10^5$ |
| Discount factor $\gamma$ | GPOMDP | 0.99 | 0.999 | 0.99 |
| | SVRPG | 0.999 | 0.999 | 0.995 |
| | SRVR-PG | 0.995 | 0.999 | 0.995 |
| Learning rate $\eta$ | GPOMDP | 0.005 | 0.005 | 0.01 |
| | SVRPG | 0.0075 | 0.0025 | 0.01 |
| | SRVR-PG | 0.005 | 0.0025 | 0.01 |
| Batch size $N$ | GPOMDP | 10 | 10 | 250 |
| | SVRPG | 25 | 10 | 250 |
| | SRVR-PG | 25 | 10 | 250 |
| Batch size $B$ | GPOMDP | - | - | - |
| | SVRPG | 10 | 5 | 50 |
| | SRVR-PG | 5 | 3 | 50 |
| Epoch size $m$ | GPOMDP | - | - | - |
| | SVRPG | 3 | 2 | 1 |
| | SRVR-PG | 3 | 2 | 1 |

Table 3: Parameters used in the SRVR-PG-PE experiments.

| Parameters | Cartpole | Mountain Car | Pendulum |
|---|---|---|---|
| NN size | - | 64 | $8\times8$ |
| NN activation function | Tanh | Tanh | Tanh |
| Task horizon | 100 | 1000 | 200 |
| Total trajectories | 2000 | 500 | 1750 |
| Discount factor $\gamma$ | 0.99 | 0.999 | 0.99 |
| Learning rate $\eta$ | 0.01 | 0.0075 | 0.01 |
| Batch size $N$ | 10 | 5 | 50 |
| Batch size $B$ | 5 | 3 | 10 |
| Epoch size $m$ | 2 | 1 | 2 |

