# OpenReview forum: "Sample Efficient Policy Gradient Methods with Recursive Variance Reduction"
_ICLR.cc/2020/Conference — Accept (Poster)_

### Official Review · AnonReviewer1 · 2019-10-17
**Official Blind Review #1**

**Rating:** 6

**Review:**

Summary:

The paper proposed a policy gradient method called SRVR-PG, which based on stochastic recursive gradient estimator. It shows that the complexity is better than that of SVRPG. Some experiments on standard environments are provided to show the efficiency of the algorithm over GPOMDP and SVRPG.

Comments:

1) I think you may ignore the highly-related work: Yang and Zhang, "Policy Optimization with Stochastic Mirror Descent", June 2019. Since both papers are highly-related, I would suggest the author(s) have some discussions to differentiate two papers.

2) Could you please provide the reasons why you are only choosing two methods (GPOMDP and SVRPG) to compare? There are many policy gradient algorithms such as A3C, A2C, ... which have not mentioned or discussed in this paper. Are they completely different here? Is there no way to compare the performance among them with SRVR-PG?

3) I am not sure if you are using the right word "novel" to describe your method. Basically, you adopt the existing estimator based on SARAH/SPIDER in optimization algorithms into RL problems. I do not think the word "novel" is proper here since it is not something total new. Notice that, the complexity result achieved in this paper is also matched the one for SARAH/SPIDER/SpiderBoost in nonconvex optimization.

4) There are some discussion on choosing a batch-size in the experimental part. However, I do not see the discussion on the learning rate. The choice of parameters for GPOMDP and SVRPG may also need to be discussed.

5) In Papini et al., 2018, for the experiment part, they use a snapshot policy to sample in the inner loop, and use early stopping inner loop. Moreover, they also check variance to recover the backup policy when it is blowup. Do you apply any trick to your experiments? I wonder if your numerical experiments are totally followed on your theory.

Minor comments:
- Redundancy ")" in \eta*S*m in Theorem 4


**Experience Assessment:**

I have published one or two papers in this area.

**Review Assessment: Checking Correctness Of Derivations And Theory:**

I carefully checked the derivations and theory.

**Review Assessment: Checking Correctness Of Experiments:**

I carefully checked the experiments.

**Review Assessment: Thoroughness In Paper Reading:**

I read the paper at least twice and used my best judgement in assessing the paper.

---

> ### Author Response · Authors · 2019-11-12
> **Response to review #1 (Part 2)**
>
> Q4: "There are some discussion on choosing a batch-size in the experimental part. However, I do not see the discussion on the learning rate. The choice of parameters for GPOMDP and SVRPG may also need to be discussed. "
>
> A4: Thank you for your suggestion. According to Theorem 4.5, the learning rate $\eta\leq 1/(4L)$, which can be chosen as a constant. In practice, we set $\eta =  C_0/L$, where $L=MR/(1-\gamma)^2$ is a constant and $C_0$ is a tuning parameter. In our experiment, for simplicity, we  use grid search to directly tune $\eta$. For instance, we searched $\eta$ for the Cartpole problem by evenly dividing the interval $[10^{-5},10^{-1}]$ into $20$ points in the log-space. The parameters of GPOMDP and SVRPG are tuned in a similar way. That is, we used the suggested values in Papini et al. (2018) and Xu et al. (2019) and multiplied them with some constants found by grid search. The detailed parameters of all methods in different environments are presented in Tables 2 and 3 in Appendix E. We have rephrased the description in Section 5 and Appendix E in the revision.
>
> Q5: "In Papini et al., 2018, for the experiment part, they use a snapshot policy to sample in the inner loop, and use early stopping inner loop ... Do you apply any trick to your experiments? I wonder if your numerical experiments are totally followed on your theory."
>
> A5: Following the source code provided by Papini et al. (2018), we normalized the gradient to avoid the gradient blowup. But we did not use any other tricks such as checking the variance to recover backup policy and the early stopping inner loop (varying epoch length) as Papini et al. (2018) did. As we explained in previous response (A.4), we set the order of batch size parameters as the theorem and corollary suggested and we tuned a constant parameter by grid search. The learning rates and other parameters are also tuned by grid search. So our numerical experiments followed our theory except the normalized gradient which is also used in Papini et al. (2018).
>
> Thank you for pointing out the typo. We have fixed it in the revision.

---

> ### Author Response · Authors · 2019-11-12
> **Response to review #1 (Part 1)**
>
> Thank you for the detailed comments. We address them point by point as follows.
>
> Q1: "I think you may ignore the highly related work ... I would suggest the author(s) have some discussions to differentiate two papers"
>
> A1: Thank you for pointing out this related work. We were not aware of it at the time of submission. In the revision, we have commented on their paper in the "Additional Related Work" section. The differences between Yang and Zhang (2019) and our paper are summarized as follows:
>
> (a) The first version (released in June) of Yang and Zhang (2019) only achieved an $O(1/\epsilon^2)$ sample complexity, which is the same as SVRPG in Papini et al. (2018). In contrast, our paper achieves $O(1/\epsilon^{3/2})$ sample complexity, which is better than theirs. Their second version was recently updated *after* the ICLR submission deadline, which also achieves $O(1/\epsilon^{3/2})$ sample complexity. Therefore, our work is concurrent and independent of their work.
>
> (b) Our algorithm is different from that of Yang and Zhang (2019). In particular, our algorithm uses the importance sampling weight to deal with the non-stationarity of the data distribution in reinforcement learning while the algorithm in Yang and Zhang (2019) does not use it. Due to our use of the step-wise importance sampling technique (equation (3.2) and Line 9 in Algorithm 1 in our paper), we prove that the smooth parameter $L$ of the performance function in our algorithm does not depend on the horizon length $H$, which is usually a very large number in practice and  can even be infinity in some cases. As a result, the sample complexity of our algorithm does not depend on $H$. In contrast, the sample complexity of the algorithm proved in Theorem 2 in Yang and Zhang (2019) depends linearly on the smooth parameter $L$,  which is in the order of $H^2$ (according to their Theorem 1) due to the lack of a proper importance weighting like in our paper. Therefore, the sample complexity of our algorithm is much lower than that of the algorithm in Yang and Zhang (2019).
>
> Q2: "Could you please provide the reasons why you are only choosing two methods (GPOMDP and SVRPG) to compare? ... "
>
> A2: In this work, we proposed a new variance reduced policy gradient algorithm (SRVR-PG) based on GPOMDP and SVRPG. We proved that SRVR-PG converges with a lower sample complexity than GPOMDP and SVRPG. Since these two algorithms and our SRVR-PG share similar structures and are comparable in terms of theoretical performance, we chose to compare them in experiments as well. We would also like to emphasize that the A3C and A2C algorithms you pointed are actor-critic type algorithms, which learn the policy (using the actor, i.e., policy gradient) and the value function (using the critic) at the same time. However, GPOMDP, SVRPG and SRVR-PG proposed in our paper are only policy gradient algorithms and do not use any critic in the update. In addition, A3C and A2C are specifically focused on parallel training while our algorithm is a general policy gradient method that can be modified to be applicable in different scenarios. Therefore, due to the fact that our algorithm does not use the actor-critic scheme (policy gradient plus critic approximation) and it is not designed for the parallel setting, our algorithm is not directly comparable to the algorithms (i.e., A3C, A2C) you pointed out.
>
> Q3: "I am not sure if you are using the right word novel to describe your method ..."
>
> A3: Although the SARAH/SPIDER estimator has been well studied in the optimization field, its application to reinforcement learning problems is non-trivial. In particular, in reinforcement learning, the data distribution is changing from iterate to iterate, thus the vanilla SPIDER estimator is not directly applicable in our setting. To overcome this challenge, we use importance sampling techniques (See equation (3.2) and the discussion around it in the revision) to correct the data sampled from one distribution to another distribution. Moreover, our importance weighted estimator is also different from Papini et al. (2018) and Xu et al. (2019) since we use the step-wise importance weight which leads to better sample complexity. Due to the use of the step-wise importance weighting, our result does not depend on the horizon length $H$, which is one of our main contributions. So we think our algorithm design is novel.

---

### Official Review · AnonReviewer3 · 2019-10-23
**Official Blind Review #3**

**Rating:** 8

**Review:**

This paper studies the theory of sample efficiency in reinforcement learning, which is of great importance and has a potentially large audience.

The strong points of the paper:
1. This paper proposed a new algorithm stochastic variance reduced policy gradient algorithms. This paper establishes better sample complexity compared with existing work. The key part of the proposed algorithm for variance reduction is to have step-wise importance weights to deal with the inconsistency caused by varying trajectory distribution.
2. This paper provides experimental results verifies the efficiency and effectiveness of the proposed algorithm.
3. In addition, parameter-based exploration extension is discussed in the appendix, which enjoys the same order of sample complexity under mild assumptions and gives better empirical performance.
4. This paper is easy to follow. In particular, there are a lot of discussions comparing this work with existing work.

The weak points of the paper:
1. In section 3, it is not quite clear how the reference policy is defined, and the \theta^s is not clearly defined when s >= 1.
2. In the main part of the paper, the discussion in Remark 4.6 and the following Corollary 4.7 is not quite clear.


Some minor comments of the paper:
1. In introduce page 3, We note that a recent work by .... by a fator of H. --> Here H should be defined as the Horizon.
2. There is one additional parenthesis in Theorem 4.5.
3. In  Corollary 4.7, T is not defined.


**Experience Assessment:**

I have read many papers in this area.

**Review Assessment: Checking Correctness Of Derivations And Theory:**

I assessed the sensibility of the derivations and theory.

**Review Assessment: Checking Correctness Of Experiments:**

I assessed the sensibility of the experiments.

**Review Assessment: Thoroughness In Paper Reading:**

I read the paper at least twice and used my best judgement in assessing the paper.

---

> ### Author Response · Authors · 2019-11-12
> **Response to review #3**
>
> Thank you for the helpful comments, which we address as follows.
>
> Q1: "In section 3, it is not quite clear how the reference policy is defined, and the $\theta^s$ is not clearly defined when $s\geq1$.
> "
>
> A1: Thanks for pointing out the missing definition. In the initialization, the reference policy is set to be the initial policy $\tilde\theta^0=\theta_0$. For $s=0,1,\ldots$, at the end of the $s$-th epoch, we update the reference policy as  $\tilde\theta^{s+1}=\theta_{m}^{s+1}$, where $\theta_{m}^{s+1}$ is the last iterate of the $s$-th epoch. We have rewritten this part in Line 3  and Line 12 of Algorithm 1 in the revision. We did not use the notation $\theta^s$ for $s\geq 1$ in our paper. We guess you are referring to $\theta_0^{s+1}$ in Line 3 of Algorithm 1 (the initial behavior policy at the beginning of each epoch), which is also set to equal to the reference policy $\tilde\theta^{s+1}$ (see Line 3 of Algorithm 1).
>
> Q2: "In the main part of the paper, the discussion in Remark 4.6 and the following Corollary 4.7 is not quite clear."
>
> A2: We have rephrased the discussion in Remark 4.6 and Corollary 4.7 in the revision. To summarize, in Remark 4.6, we compare our convergence rate with that in Papini et al. (2018) and show that our result does not have the additional additive term $1/B$ where $B$ is the batch size in inner loops. We also compare our results with that in Xu et al. (2019) and show that our batch size $B$ does not depend on the horizon length $H$, which is another big improvement.
>
> In Corollary 4.7 and the discussion after it, we calculate the total number of samples (trajectories) that are required by our algorithm in order to find an $\epsilon$-approximate stationary point. We show that our sample complexity is lower than that of Xu et al. (2019) by a factor of $O(1/\epsilon^{1/6})$.
>
>
> Thank you for pointing out the typos. We have fixed them in the revision. We have removed the notation $T$ since it can be replaced by $T=Sm$ which makes the dependency clearer.  We added the choice of $S=O(1/\epsilon^{1/2})$ in Corollary 4.7.

---

### Official Review · AnonReviewer2 · 2019-10-24
**Official Blind Review #2**

**Rating:** 6

**Review:**

The authors propose a new stochastic reduced variance policy gradient estimator, which combines a baseline GPOMDP estimator with a control variate integrating past gradients by importance re-weighting. The authors establish the sample complexity of gradient descent using the proposed estimator, and further demonstrate its effectiveness through some simple empirical results.

I believe this paper is a good contribution for ICLR. The result is relevant and interesting, and extends recent ideas around reduced-variance policy gradient estimators. The paper is overall easy to read, and presents its ideas clearly. Some detailed comments:

- The wording of theorem 4.5 and corollary 4.7 could be somewhat clarified. In particular, I did not see \Phi defined in the main text, and given its definition in the appendix, I believe theorem 4.5 could be stated simply in terms of J, avoiding any additional notation. Similarly, corollary 4.7 could be stated somewhat more clearly, and in particular, the choice of S should be made explicit. In the appendix, I could not find a definition for \phi.

- The empirical results presented are interesting, although I wish they were more comprehensive. In particular, it would be valuable to more exhaustively evaluate the impact of the hyper-parameters N, B and m. The authors should also clarify how the current values were chosen. Given that the theoretical results also apply to projected gradient descent, it would be interesting to see empirical results in that case.

**Experience Assessment:**

I have read many papers in this area.

**Review Assessment: Checking Correctness Of Derivations And Theory:**

I did not assess the derivations or theory.

**Review Assessment: Checking Correctness Of Experiments:**

I assessed the sensibility of the experiments.

**Review Assessment: Thoroughness In Paper Reading:**

I read the paper at least twice and used my best judgement in assessing the paper.

---

> ### Author Response · Authors · 2019-11-12
> **Response to review #2**
>
> Thank you for your constructive comments. We respond to your concerns point by point as follows.
>
> Q1a: "The wording of theorem 4.5 and corollary 4.7 could be somewhat clarified. In particular, I did not see $\Phi$ defined in the main text, and given its definition in the appendix, I believe theorem 4.5 could be stated simply in terms of J, avoiding any additional notation."
>
> A1a: Thanks for the suggestion. We have removed $\Phi$ in the statement of Theorem 4.5 and used $J$ and the indicator function  according to your advice.
>
> Q1b: "Similarly, corollary 4.7 could be stated somewhat more clearly, and in particular, the choice of S should be made explicit."
>
> A1b: By Theorem 4.5 and Remark 4.6, we have $Sm=T=O(1/\epsilon)$. Therefore, the choice of $S$ in Corollary 4.7 is $S=T/m=O(1/\epsilon^{1/2})$ since we have set $m=O(1/\epsilon^{1/2})$. We have added this in the statement of Corollary 4.7 in the revision.
>
> Q1c: "In the appendix, I could not find a definition for $\varphi$."
>
> A1c: We are sorry for the typo in equation (B.3) in the appendix. The function $\varphi_{\mathbf{\Theta}}$ should be the set indicator function  $\mathbf{1}_{\mathbf{\Theta}}$ over the set $\mathbf{\Theta}$. We have replaced $\varphi_{\mathbf{\Theta}}$ by the set indicator function $\mathbf{1}_{\mathbf{\Theta}}$ in equation (B.3) in the revision.
>
> Q2a: "The empirical results presented are interesting, although I wish they were more comprehensive. In particular, it would be valuable to more exhaustively evaluate the impact of the hyper-parameters N, B and m."
>
> A2a: Thank you for the suggestions. According to the proof of Corollary 4.7 in our paper (please see Appendix B), in order to achieve $\epsilon$ precision, $N$ should be in the order of $1/\epsilon$. Since we have $mB=N$ and $N=1/\epsilon$, when we change the scale of $B$, $m$ will be changed accordingly as $m = N/B$. Therefore, we only need to evaluate the selection of hyper-parameter $B$ in our experiment, because once $B$ is chosen, $m$ is determined accordingly."
>
> Q2b: "The authors should also clarify how the current values were chosen."
>
> A2b: According to Corollary 4.7, the orders of the batch size parameters $N$ and $B$ and the epoch length $m$ should be chosen based on the user-defined precision parameter $\epsilon$. In particular, we require $N=O(1/\epsilon)$ and $mB=O(1/\epsilon)$. Based on our theoretical result in Corollary 4.7 and our sensitivity study on the inner loop batch size $B$, we further choose $B=O(1/\epsilon^{1/2})$ and thus $m=O(1/\epsilon^{1/2})$. In our experiments, we set $N=C_0/\epsilon$, $B=C_1/\epsilon^{1/2}$ and $m=C_2/\epsilon^{1/2}$ and tune the constant parameters $C_0, C_1, C_2$ using grid search.
>
> Q2c: "Given that the theoretical results also apply to projected gradient descent, it would be interesting to see empirical results in that case."
>
> A2c: The projection step is only employed in the theoretical analysis to cover cases where the optimal policy is constrained in certain convex set. In our experiments, we found that the proposed algorithm works well without the extra projection step for all the environments considered in our paper. Therefore, we did not use projection in our experiments.

---

### Comment · Area_Chair1 · 2019-11-13
**Thanks for your reviews. Please take a look at the rebuttal.**

Dear reviewers,

Thank you very much for your efforts in reviewing this paper.

The authors have provided their rebuttal. It would be great if you take a look at them, and see whether it changes your opinion in anyway. If there is still any unclear point or a serious disagreement, please bring it up. Also if you are hoping to see a specific change or clarification in the paper before you update your score, please mention it.

The authors have only until November 15th to reply back.

I also encourage you to take a look at each others’ reviews. There might be a remark in other reviews that changes your opinion.

Thank you,
Area Chair

---

### Decision · Program_Chairs · 2019-12-19

**Decision:**

Accept (Poster)

**Comment:**

The paper introduces a policy gradient estimator that is based on stochastic recursive gradient estimator. It provides a sample complexity result of O(eps^{-3/2}) trajectories for estimating the gradient with the accuracy of eps.
This paper generated a lot of discussions among reviewers. The discussions were around the novelty of this work in relation to SARAH (Nguyen et al., ICML2017), SPIDER (Fang et al., NeurIPS2018) and the work of Papini et al. (ICML 2018). SARAH/SPIDER are stochastic variance reduced gradient estimators for convex/non-convex problems and have been studied in the optimization literature.
To bring it to the RL literature, some adjustments are needed, for example the use of importance sampling (IS) estimator. The work of Papini et al. uses IS, but does not use SARAH/SPIDEH, and it does not use step-wise IS.

Overall, I believe that even though the key algorithmic components of this work have been around, it is still a valuable contribution to the RL literature.